# Generalizable and Animatable Gaussian Head Avatar

**Xuangeng Chu**
The University of Tokyo
`xuangeng.chu@mi.t.u-tokyo.ac.jp`

**Tatsuya Harada**
The University of Tokyo
RIKEN AIP
`harada@mi.t.u-tokyo.ac.jp`

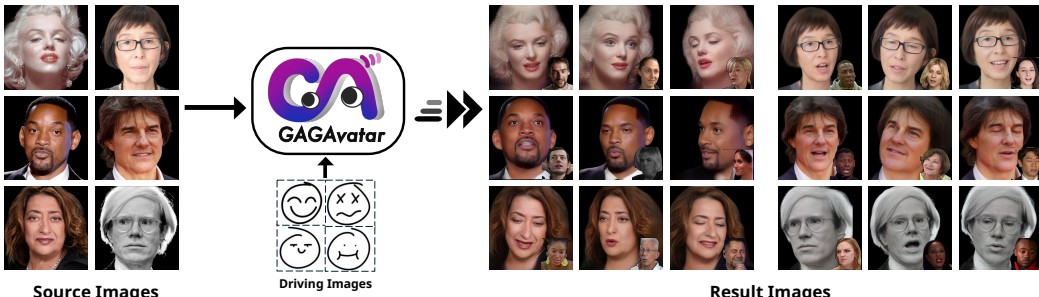

Figure 1: Our method can reconstruct animatable avatars from a single image, offering strong generalization and controllability with real-time reenactment speeds.

## Abstract

In this paper, we propose **G**eneralizable and **A**nimatable **G**aussian head **A**vatar (GAGAvatar) for one-shot animatable head avatar reconstruction. Existing methods rely on neural radiance fields, leading to heavy rendering consumption and low reenactment speeds. To address these limitations, we generate the parameters of 3D Gaussians from a single image in a single forward pass. The key innovation of our work is the proposed dual-lifting method, which produces high-fidelity 3D Gaussians that capture identity and facial details. Additionally, we leverage global image features and the 3D morphable model to construct 3D Gaussians for controlling expressions. After training, our model can reconstruct unseen identities without specific optimizations and perform reenactment rendering at real-time speeds. Experiments show that our method exhibits superior performance compared to previous methods in terms of reconstruction quality and expression accuracy. We believe our method can establish new benchmarks for future research and advance applications of digital avatars. Code and demos are available at https://github.com/xg-chu/GAGAvatar.

## 1 Introduction

One-shot head avatar reconstruction has garnered significant attention in computer vision and graphics recently due to its great potential in applications such as virtual reality and online meetings. The typical problem involves faithfully recreating the source head from one image while precisely controlling expressions and poses. In recent years, many exploratory methods have achieved this goal using 2D generative models and 3D synthesizers.

Some early 2D-based methods [Yin et al., 2022, Ren et al., 2021] typically combine estimated deformation fields with generative networks to drive images. However, due to the lack of necessary 3D constraints and modeling, these methods struggle to maintain multi-view consistency of expressions and identities when head poses change significantly. Recently, Neural Radiance Fields (NeRF) [Mildenhall et al., 2020] have shown impressive results in head avatar synthesis, providing

38th Conference on Neural Information Processing Systems (NeurIPS 2024).

solutions using 3D synthesizers to achieve realistic details such as accessories and hair. However, some NeRF-based methods [Ma et al., 2023] require identity-specific training and optimization, and some methods [Li et al., 2023a, Chu et al., 2024, Deng et al., 2024a] can't render in real-time during inference, limiting their application in certain scenarios. With the emergence of 3D Gaussian splatting [Kerbl et al., 2023], some methods [Xu et al., 2024] have achieved real-time rendering. However, these methods still require specific training for each identity and fail to generalize to unseen identities, leaving the modeling of generalizable 3D Gaussian-based head models unexplored.

To address these limitations, we introduce a novel 3D Gaussian-based framework for one-shot head avatar reconstruction. Given a single image, our framework reconstructs an animatable 3D Gaussian-based head avatar, achieving real-time expression control and rendering. Some examples are shown in Fig. 1. The core challenge lies in faithfully reconstructing 3D Gaussians from a single image, as a 3D Gaussian typically requires multi-view input and millions of Gaussian points for detailed reconstruction. To address this, we propose a novel dual-lifting method that reconstructs the 3D Gaussians from one image. Specifically, instead of directly estimating Gaussian points from the image, we predict the lifting distances of each pixel relative to the image plane, and then map the image plane and lifted points back to 3D space based on the camera position. By predicting forward and backward lifting distances, we can form an almost closed Gaussian points distribution and reconstruct the head as completely as possible. This approach leverages the fine-grained features of the input image and significantly reduces the difficulty of predicting 3D Gaussian positions. We also utilize priors from 3D Morphable Models (3DMM) [Li et al., 2017] to further constrain the lifting distance, helping the model obtain correct 3D lifting and capture details from the source image. We then bind learnable features to the 3DMM vertices and construct expression Gaussians using image global features, 3DMM learnable features, and 3DMM point positions to ensure expression control capability. Finally, we use a neural renderer to refine the splatting-rendered results, producing the final reenacted image. Our model is learned from a large number of monocular portrait images and can be generalized to unseen identities after training. Experiments verify that our method performs better than previous methods in terms of reconstruction quality and expression accuracy, and achieves real-time reenactment and rendering speed.

Our major contributions can be summarized as follows:

- We propose GAGAvatar, which to our knowledge is the first generalizable 3D Gaussian head avatar framework that achieves single forward reconstruction and real-time reenactment.

- To achieve this, we propose a dual-lifting method to lift Gaussians from a single image and introduce a method that uses 3DMM priors to constrain the lifting process.

- We combine 3DMM priors and 3D Gaussians to accurately transfer expression information while avoiding redundant computations.

## 2 Related Work

### 2.1 2D-based Talking Head Synthesis

The impressive performance of CNN and Generative Adversarial Networks (GAN) [Goodfellow et al., 2014, Isola et al., 2017, Karras et al., 2020] has inspired many methods for direct head image synthesis using 2D networks. A popular strategy of early works is inserting the expression and head pose features of the driving image into the 2D generative network to achieve realistic and animatable image generation. For example, these methods [Zakharov et al., 2019, Burkov et al., 2020, Zhou et al., 2021, Wang et al., 2023] inject latent representations of expression into the U-Net backbone or StyleGAN-like [Karras et al., 2019] generators to transfer driving expressions to reenacted images. A recent trend in 2D-based talking head synthesis methods [Siarohin et al., 2019, Ren et al., 2021, Drobyshev et al., 2022, Hong et al., 2022a, Zhang et al., 2023a] is to represent expressions and head poses as warp fields, performing expression transfer by deforming the source image to match the driving image. However, due to the lack of explicit understanding of the 3D geometry of head portraits, these methods often produce unrealistic distortions and undesired identity changes when there are significant pose and expression variations. Although some methods [Drobyshev et al., 2022, Wang et al., 2021a, Ren et al., 2021, Yin et al., 2022, Zhang et al., 2023b] introduce 3D Morphable Models (3DMM) [Blanz and Vetter, 1999, Paysan et al., 2009, Li et al., 2017, Gerig et al., 2018]

into the 2D framework, they still lack the ability to control the viewpoint and achieve free-viewpoint rendering. Additionally, there are some audio-driven 2D control methods [Guo et al., 2021, Tang et al., 2022, Zhang et al., 2023b], while flexible to use, cannot explicitly control facial expressions and poses, sometimes resulting in unsatisfactory outcomes. In contrast, our method uses an explicit 3D representation to enable free view control and realistic synthesis even under large pose variations.

## 2.2 3D-based Head Avatar Reconstruction

To achieve better 3D consistency in head avatars, many works have explored using 3D representations for reconstruction. Early methods [Xu et al., 2020, Khakhulin et al., 2022] used 3DMM-based meshes [Li et al., 2017, Gerig et al., 2018] to reconstruct head avatars. Since neural radiance fields (NeRF) [Mildenhall et al., 2020] have demonstrated excellent results, many recent methods [Li et al., 2023b,a, Ma et al., 2023, Yu et al., 2023, Chu et al., 2024, Ye et al., 2024, Deng et al., 2024b,a, Park et al., 2021a, Zheng et al., 2023, Bai et al., 2023a, Ki et al., 2024] have adopted NeRF for head reconstruction. However, some approaches [Gafni et al., 2021, Park et al., 2021a, Tretschk et al., 2021, Hong et al., 2022b, Athar et al., 2022, Park et al., 2021b, Gao et al., 2022, Guo et al., 2021, Bai et al., 2023b, Kirschstein et al., 2023, Zheng et al., 2023, Bai et al., 2023a, Zhao et al., 2023, Zhang et al., 2024] require multi-view or single-view videos of specific identities for training, limiting generalization and raising privacy concerns due to the need for thousands of frames of personal image data. Additionally, some methods [Xu et al., 2023a, Tang et al., 2023, Sun et al., 2022, Xu et al., 2023b, Zhuang et al., 2022a, Sun et al., 2023] train generators to produce controllable head avatars from random noise, followed by inversion [Roich et al., 2022, Xie et al., 2023] for identity-specific reconstruction. These methods often suffer from inversion accuracy limitations, failing to preserve the identity of the source image. There are also methods [Hong et al., 2022b, Zhuang et al., 2022b, Ma et al., 2023] to perform test-time optimization on the source image to obtain reconstructions, but the need for test-time optimization limits their applicability. To address these challenges, some works [Yu et al., 2023, Li et al., 2023a,b, Ma et al., 2024a, Yang et al., 2024, Chu et al., 2024, Ye et al., 2024, Ma et al., 2024a, Deng et al., 2024b,a] focus on one-shot head reconstruction without test-time optimization. For example, GOHA [Li et al., 2023a] learns three tri-plane features to capture details. HideNeRF [Li et al., 2023b] utilizes multi-resolution tri-planes and a deformation field to generate reenactment images. GPAvatar [Chu et al., 2024] uses a point-based expression field and a multi tri-plane attention module to reconstruct head avatars. Real3DPortrait [Ye et al., 2024] generates a tri-plane from images and adds motion adapters to get reenactment images. CVTHead [Ma et al., 2024a] reconstructs head avatars using point-based neural rendering and a vertex-feature transformer. Portrait4D [Deng et al., 2024b] learns dynamic expression tri-plane from multi-view synthetic data, while Portrait4D-v2 [Deng et al., 2024a] learns from pseudo multi-view videos, addressing the lack of real video training in Portrait4D. However, these NeRF-based methods often face rendering speed limitations, preventing real-time application. Methods [Xu et al., 2024, Li et al., 2024, Hu et al., 2023, Wang et al., 2024a, Ma et al., 2024b, Wang et al., 2024b] utilizing 3D Gaussian splatting[Kerbl et al., 2023] achieve excellent performance and rendering speed but require video data for identity-specific training, lacking generalization capabilities. In this paper, we propose a one-shot 3D Gaussian head avatar reconstruction method based on the dual-lifting method. Our method can generalize to unseen identities, achieves real-time rendering, and surpasses previous works in image quality.

## 3 Method

An overview of the reenactment process of our method is shown in Fig. 2. Given a source image $I_s$, we first use DINOv2 [Darcet et al., 2023, Oquab et al., 2023] to extract global and local features. Using the local features, we apply our proposed dual-lifting methods to predict the parameters and positions of two 3D Gaussians. Simultaneously, we assign learnable parameters to each vertex of the 3DMM [Li et al., 2017] model and predict another expression Gaussians using the combination of the global feature and vertex features. We directly use the vertex positions of the 3DMM model as the positions for expression Gaussians. Finally, we combine these 3D Gaussians and perform splatting to produce a coarse result image $I_c$ with the expression and pose of driving image $I_d$, which is then further refined through a neural renderer to obtain the fine result image $I_f$.

In the following subsections, we describe the reconstruction branch based on dual-lifting in Sec. 3.1, explain the expression modeling and control branch in Sec. 3.2, and detail our neural renderer in Sec. 3.3. Finally, we describe our lifting distance loss and the training objectives in Sec. 3.4.

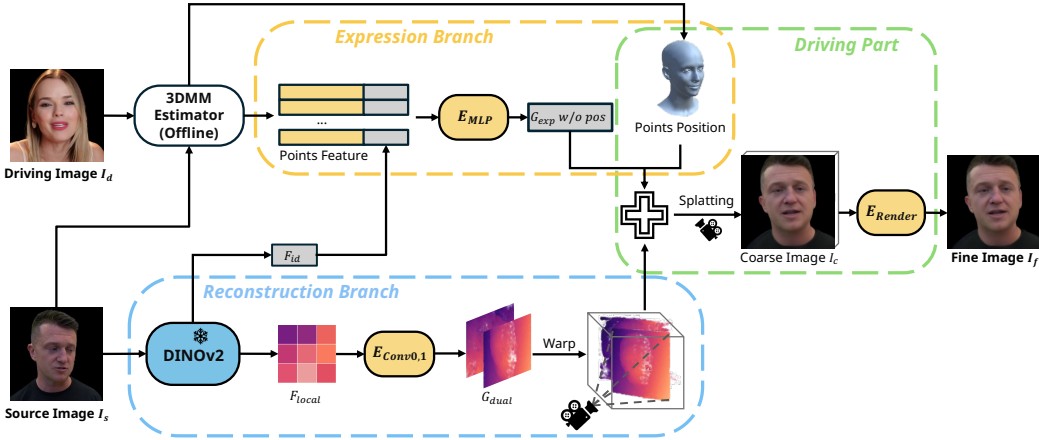

Figure 2: Our method consists of two branches: a reconstruction branch (Sec. 3.1) and an expression branch (Sec. 3.2). We render dual-lifting and expressed Gaussians to get coarse results, and then use a neural renderer to get fine results. Only a small driving part needs to be run repeatedly to drive the expression, while the rest is executed only once.

## 3.1 Dual-lifting and Reconstruction Branch

Given an input source image, our goal is to reconstruct a detailed 3D head avatar. To ensure stable modeling and learning, we impose certain constraints on the reconstruction process. First, we assume that the reconstructed head is always located at the origin in normalized 3D space. Second, the rotation of the head is modeled through changes in camera pose to ensure that the head itself is relatively stationary. We follow the same strategy when tracking 3DMM parameters and camera parameters from training and testing data. These constraints allow the model to effectively utilize the stable priors of the human head topology.

Leveraging the success of 3D Gaussians splatting [Kerbl et al., 2023] in synthesis quality and rendering speed, we propose a dual-lifting method to reconstruct 3D Gaussians from a single image. Reconstructing 3D Gaussians typically requires millions of points, but obtaining such a dense density of Gaussian points from a single image is a challenging task, especially without test-time optimization. To address this problem, we propose a novel reconstruction method: the dual-lifting method. Briefly, we first get the local feature plane $F_{local}$ by a frozen DINOv2 backbone, and then predict the offsets of each pixel relative to the feature plane and the other necessary parameters (including color, opacity, scale and rotation), instead of predicting the 3D Gaussians directly. We then map the plane back to 3D space based on the camera pose and place the plane through the origin, which provides the 3D position and normal vector of the plane pixels. Finally, we can calculate the position of these 3D Gaussians in 3D space based on the predicted offsets, positions and normal vector. This process can be described as follows:

$$G_{pos} = [p_s + E_{Conv0}(F_{local}) \cdot n_s, \quad p_s - E_{Conv1}(F_{local}) \cdot n_s], \tag{1}$$

$$G_{c,o,s,r} = [E_{Conv0}(F_{local}), \quad E_{Conv1}(F_{local})], \tag{2}$$

where $p_i$ is the initial points plane mapped based on the estimated camera pose of $I_s$ and passes through the origin. The size of $p_i$ is $296 \times 296$, which is consistent with the local feature $F_{local}$. $E_{Conv0,1}$ are convolutional networks, $n_s$ is the normal vector of $p_s$, $G_{pos}$ is the position of reconstructed 3D Gaussians, and $G_{c,o,s,r}$ represents the color, opacity, scale, and rotation of 3D Gaussians.

It's worth noting that while predicting one set of lifting distances from the plane is possible, we adopted a strategy of predicting forward and backward lifting separately. Our dual-lifting method aims to predict a complete 3D structure from a single source image, to achieve multi-view consistency during inference. If we predict only one set of lifting distances from the image plane, we may face some ambiguous situations during learning. For example, when we want to reconstruct a side view source image, predicting one set of lifting will simultaneously lift the point forward to the visible surface and backward to include the other side of the head. During this process, each pixel can be lifted to the visible surface or to the opposite surface, as both are justified, resulting in model

performance degradation. Unlike single-lifting prediction, our dual-lifting strategy predicts forward and backward lifting separately, which eliminates ambiguities and stabilizes the optimization process.

Our dual-lifting method effectively exploits the detailed information of the source image to reconstruct 3D Gaussians. At the same time, the two sets of predicted lifting points can form an almost closed Gaussian points distribution, thus enhancing the performance of large viewpoint changes. The 3D Gaussian generated by dual-lifting can be rendered from any viewpoint, producing static results. In the next section, we describe how to control the facial expressions of the generated avatar.

## 3.2 Expression Branch

Expression transfer is not a straightforward task, but the 3DMM [Li et al., 2017] provides us with a powerful tool to represent common facial expressions and decouple expressions from identity, thereby facilitating expression control. Our expression branch establishes 3D Gaussians based on the 3DMM vertices to control the expressions of the generated images. To achieve this, we bind learnable weights to each vertex in the 3DMM. Due to the stable semantics of 3DMM vertices, the features of these points correspond to facial positions such as the eyes and mouth.

As shown in Fig. 2, given the source image $I_s$ and driving image $I_d$, we concatenate the global features $F_{id}$ with the learnable features of vertices. We then use a MLP to predict the Gaussian parameters (excluding position) of each point from these features, and use the position of the 3DMM vertices. Here we combine the global features $F_{id}$ of the source image when predicting the expression Gaussians. This will introduce identity information to the expression branch and enhance the identity consistency under various expressions, as confirmed by our experiments. Throughout the driving process, we only need to infer the Gaussians of the reconstruction branch and expression branch once. Reenactment is achieved by modifying the camera pose and position of the Gaussians in the expression branch, which allows us to perform fast reenactment without redundant calculations.

## 3.3 Neural Renderer

Reconstructing 3D Gaussians typically requires millions of points, but in our dual-lifting method, we generate only 175,232 points. These Gaussians can reconstruct the target avatar, but with RGB information alone it is insufficient for capturing the rich details of a human avatar. To enhance the representation capability of the sparse Gaussians, we predict 32-dimensional features containing RGB information and then perform splatting to obtain coarse images. Then we use a popular neural renderer following existing methods [Li et al., 2023a, Chu et al., 2024, Ye et al., 2024] to get the fine image, as Fig. 2 shows. Unlike these methods which use neural render as a super-resolution module to reduce rendering time, we do not upsample the image as our method do not face significant rendering time issues. Our neural renderer effectively decodes the dual lifting and expression Gaussians features into RGB values, producing high-quality results and resolving potential conflicts between the two sets of Gaussians. We train our neural renderer from scratch during the training process, without any pre-trained initialization.

## 3.4 Training Strategy and Loss Functions

With the exception of the frozen DINOv2 backbone, we train the model from scratch. During training, we randomly sample two images from the same video, one as the source image and the other as the driving image and target image. Our primary objective during training is to ensure that the reenacted coarse and fine image aligns with the target image. Given that both images share the same identity, this alignment is achievable. We employ L1 loss and perceptual loss [Johnson et al., 2016, Zhang et al., 2018, Ye et al., 2024] on both the coarse and the fine image.

Additionally, we propose a lifting distance loss $\mathcal{L}_{lifting}$ to assist dual-lifting learning. With the help of the prior provided by the tracked 3DMM, we require the lifting distance predicted by the network to be as close as possible to the 3DMM vertices. Specifically, we look for the lifting point closest to each 3DMM vertex and constrain their distance through L2 loss. The calculation is as follows:

$$\mathcal{L}_{lifting} = ||P_{3dmm} - \{argmin_{q \in G_{pos}} ||p - q|| \mid p \in P_{3dmm}\} ||, \qquad (3)$$

where the $P_{3dmm}$ is the tracked 3DMM vertices, $G_{pos}$ is the dual-lifting points, $argmin$ find the nearest point. Our lifting distance loss leverages 3DMM priors. Additionally, since we constrain only

| Source | Driving | GOHA | CVTHead | GPAvatar | Real3DPotrait | Potrait4D | Potrait4Dv2 | Ours |
|--------|---------|------|---------|----------|---------------|-----------|-------------|------|

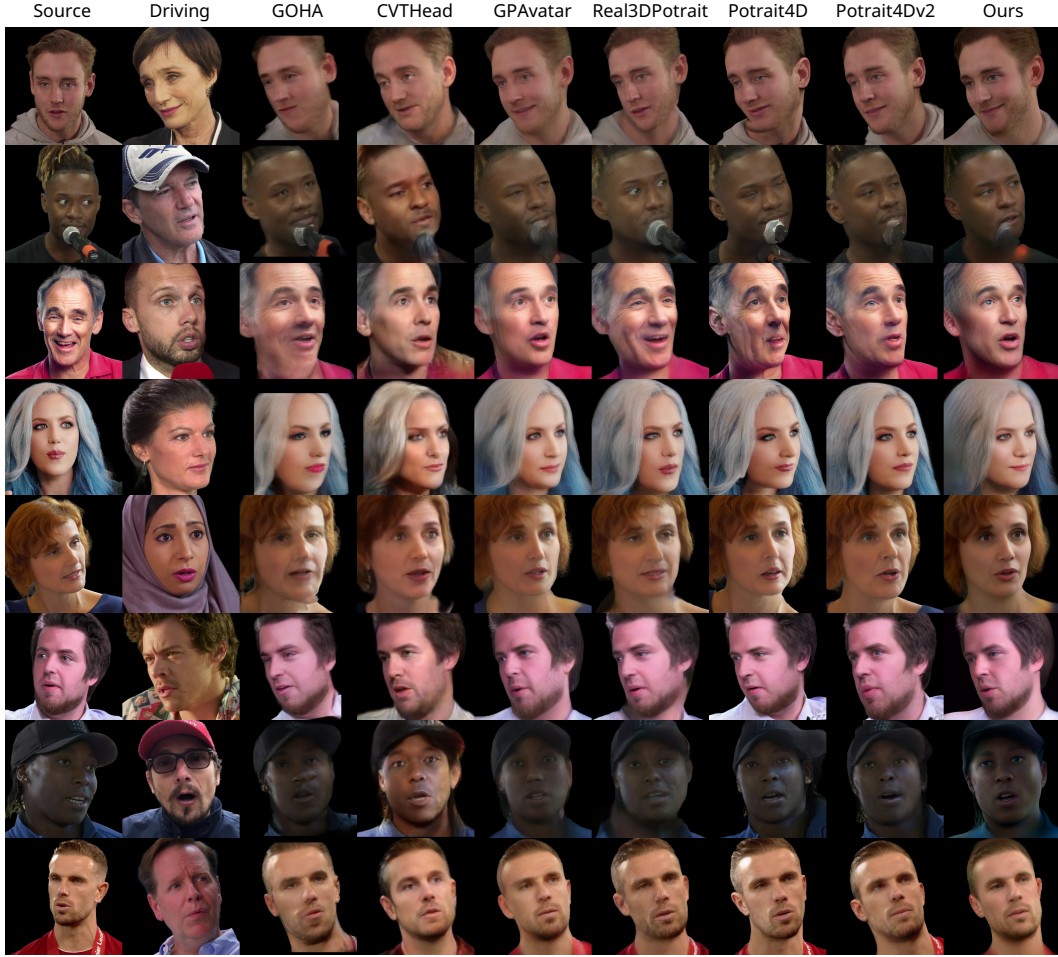

Figure 3: Cross-identity qualitative results on the VFHQ [Xie et al., 2022] dataset. Compared with baseline methods, our method has accurate expressions and rich details.

a subset of dual-lifting points, the model can still learn areas not modeled by 3DMM, such as hair and accessories. Experiments show $\mathcal{L}_{lifting}$ can improve the 3D structure and the performance of large view changes.

The overall training objective is as follows:

$$\mathcal{L} = ||I_c - I_t|| + ||I_f - I_t|| + \lambda_p(||\varphi(I_c) - \varphi(I_t)|| + ||\varphi(I_f) - \varphi(I_t)||) + \lambda_l\mathcal{L}_{lifting}, \quad (4)$$

where $I_t$ is target image, $I_c$ and $I_f$ are the generated coarse and fine image, $\lambda_p$ and $\lambda_l$ are the weights used to balance the losses.

## 4 Experiments

### 4.1 Experiment Setting

**Datasets.** We use the VFHQ [Xie et al., 2022] dataset to train our model, which comprises clips from various interview scenarios. To avoid consecutive similar frames, we sampled 25 to 75 frames from the original video depending on video length. This resulted in a dataset that includes 586,382 frames from 15,204 video clips. All the images are resized to 512×512. We tracked camera poses, FLAME [Li et al., 2017] parameters and removed the background following [Chu et al., 2024]. For evaluation, we use sampled frames from the VFHQ original test split, consisting of 5000 frames from 100 videos. The first frame of each video serves as the source image, with the remaining frames used

Table 1: Quantitative results on the VFHQ [Xie et al., 2022] dataset. We use colors to denote the first, second and third places respectively.

| Method | Self Reenactment | | | | | | | Cross Reenactment | | |
|---|---|---|---|---|---|---|---|---|---|---|
| | PSNR↑ | SSIM↑ | LPIPS↓ | CSIM↑ | AED↓ | APD↓ | AKD↓ | CSIM↑ | AED↓ | APD↓ |
| StyleHeat [Yin et al., 2022] | 19.95 | 0.726 | 0.211 | 0.537 | 0.199 | 0.385 | 7.659 | 0.407 | 0.279 | 0.551 |
| ROME [Khakhulin et al., 2022] | 19.96 | 0.786 | 0.192 | 0.701 | 0.138 | 0.186 | 4.986 | 0.530 | 0.259 | 0.277 |
| OTAvatar [Ma et al., 2023] | 17.65 | 0.563 | 0.294 | 0.465 | 0.234 | 0.545 | 18.19 | 0.364 | 0.324 | 0.678 |
| HideNeRF [Li et al., 2023b] | 19.79 | 0.768 | 0.180 | 0.787 | 0.143 | 0.361 | 7.254 | 0.514 | 0.277 | 0.527 |
| GOHA [Li et al., 2023a] | 20.15 | 0.770 | 0.149 | 0.664 | 0.176 | 0.173 | 6.272 | 0.518 | 0.274 | 0.261 |
| CVTHead [Ma et al., 2024a] | 18.43 | 0.706 | 0.317 | 0.504 | 0.186 | 0.224 | 5.678 | 0.374 | 0.261 | 0.311 |
| GPAvatar [Chu et al., 2024] | 21.04 | 0.807 | 0.150 | 0.772 | 0.132 | 0.189 | 4.226 | 0.564 | 0.255 | 0.328 |
| Real3DPortrait [Ye et al., 2024] | 20.88 | 0.780 | 0.154 | 0.801 | 0.150 | 0.268 | 5.971 | 0.663 | 0.296 | 0.411 |
| Portrait4D [Deng et al., 2024b] | 20.35 | 0.741 | 0.191 | 0.765 | 0.144 | 0.205 | 4.854 | 0.596 | 0.286 | 0.258 |
| Portrait4D-v2 [Deng et al., 2024a] | 21.34 | 0.791 | 0.144 | 0.803 | 0.117 | 0.187 | 3.749 | 0.656 | 0.268 | 0.273 |
| Ours | 21.83 | 0.818 | 0.122 | 0.816 | 0.111 | 0.135 | 3.349 | 0.633 | 0.253 | 0.247 |

Table 2: Quantitative results on the HDTF [Zhang et al., 2021] dataset. We use colors to denote the first, second and third places respectively.

| Method | Self Reenactment | | | | | | | Cross Reenactment | | |
|---|---|---|---|---|---|---|---|---|---|---|
| | PSNR↑ | SSIM↑ | LPIPS↓ | CSIM↑ | AED↓ | APD↓ | AKD↓ | CSIM↑ | AED↓ | APD↓ |
| StyleHeat [Yin et al., 2022] | 21.41 | 0.785 | 0.155 | 0.657 | 0.158 | 0.162 | 4.585 | 0.632 | 0.271 | 0.239 |
| ROME [Khakhulin et al., 2022] | 20.51 | 0.803 | 0.145 | 0.738 | 0.133 | 0.123 | 4.763 | 0.726 | 0.268 | 0.191 |
| OTAvatar [Ma et al., 2023] | 20.52 | 0.696 | 0.166 | 0.662 | 0.180 | 0.170 | 8.295 | 0.643 | 0.292 | 0.222 |
| HideNeRF [Li et al., 2023b] | 21.08 | 0.811 | 0.117 | 0.858 | 0.120 | 0.247 | 5.837 | 0.843 | 0.276 | 0.288 |
| GOHA [Li et al., 2023a] | 21.31 | 0.807 | 0.113 | 0.725 | 0.162 | 0.117 | 6.332 | 0.735 | 0.277 | 0.136 |
| CVTHead [Ma et al., 2024a] | 20.08 | 0.762 | 0.179 | 0.608 | 0.169 | 0.138 | 4.585 | 0.591 | 0.242 | 0.203 |
| GPAvatar [Chu et al., 2024] | 23.06 | 0.855 | 0.104 | 0.855 | 0.114 | 0.135 | 3.293 | 0.842 | 0.268 | 0.219 |
| Real3DPortrait [Ye et al., 2024] | 22.82 | 0.835 | 0.103 | 0.851 | 0.138 | 0.137 | 4.640 | 0.903 | 0.299 | 0.238 |
| Portrait4D [Deng et al., 2024b] | 20.81 | 0.786 | 0.137 | 0.810 | 0.134 | 0.131 | 4.151 | 0.793 | 0.291 | 0.240 |
| Portrait4D-v2 [Deng et al., 2024a] | 22.87 | 0.860 | 0.105 | 0.860 | 0.111 | 0.111 | 3.292 | 0.857 | 0.262 | 0.183 |
| Ours | 23.13 | 0.863 | 0.103 | 0.862 | 0.110 | 0.111 | 2.985 | 0.851 | 0.231 | 0.181 |

as driving and target images for reenactment. We also evaluate on HDTF [Zhang et al., 2021] dataset, following the test split used in [Ma et al., 2023, Li et al., 2023a], including 19 video clips.

**Implementation details.** Our framework is built on the PyTorch [Paszke et al., 2017] platform. We use FLAME [Li et al., 2017] as our driving 3DMM. During training, we use the ADAM [Kingma and Ba, 2014] optimizer with a learning rate of 1.0e-4. The DINOv2 [Oquab et al., 2023] backbone is frozen during training and is not trained or fine-tuned. Our training consists of 200,000 iterations with a total batch size of 8. The training process is conducted on an NVIDIA Tesla A100 GPU and takes approximately 46 GPU hours, demonstrating efficient resource utilization. During inference, our method achieves 67 FPS on an A100 GPU while using only 2.5 GB of VRAM, showcasing high efficiency. Further implementation details of the model can be found in the supplementary materials.

## 4.2 Main Results

**Baseline methods.** We conduct comparisons with existing state-of-the-art methods, including ROME [Khakhulin et al., 2022], StyleHeat [Yin et al., 2022], OTAvatar [Ma et al., 2023], HideN-eRF [Li et al., 2023b], GOHA [Li et al., 2023a], CVTHead [Ma et al., 2024a], GPAvatar [Chu et al., 2024], Real3DPortrait [Ye et al., 2024], Portrait4D [Deng et al., 2024b], and Portrait4D-v2 [Deng et al., 2024a]. For each method, we use the official implementation to obtain the result. It is worth noting that actually the core contributions of Portrait4D-v2 are orthogonal to our work. They introduced a new data generation method and a novel learning paradigm to improve performance, which means our method can also benefit from their advancements.

**Qualitative results.** Fig. 3 shows qualitative comparisons between methods. Compared with other methods, our method can reconstruct detailed head avatars from source images and capture subtle facial movements such as eyes and mouth in driving images. Our method can also maintain identity

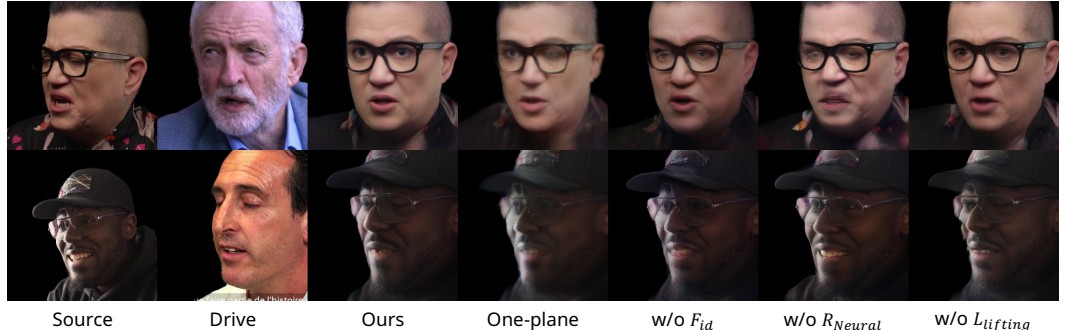

| Source | Drive | Ours | One-plane | w/o $F_{id}$ | w/o $R_{Neural}$ | w/o $L_{lifting}$ |

Figure 4: Ablation results on VFHQ [Xie et al., 2022] datasets. We can see that our full method performs best, especially on facial edges such as glasses in large view angles.

consistency and image quality when handling large head rotations. At the same time, our method achieves high-quality reconstruction and rendering at a much faster speed than the baseline method.

**Quantitative results.** We also quantitatively evaluate the self and cross-identity reenactment performance between methods. For self-reenactment with ground truth available, we measure the quality of the synthesized images using PSNR, SSIM, LPIPS [Zhang et al., 2018] between the synthetic results and the ground truth. For identity similarity, we calculate the cosine distance of face recognition features [Deng et al., 2019a] between the reenactment results and the source images. For expression and pose, we use the average expression distance (AED) and average pose distance (APD) measured by a 3DMM estimator [Deng et al., 2019b], and the average keypoint distance (AKD) based on a facial landmark detector [Bulat and Tzimiropoulos, 2017] to evaluate the accuracy of driving control. For the cross-identity reenactment task, due to the lack of ground truth, we evaluate CSIM, AED, and APD, generally consistent with previous work [Li et al., 2023a, Chu et al., 2024, Ye et al., 2024].

Tab. 1 and Tab. 2 show the quantitative results on the VFHQ and HDTF datasets, respectively. Our method outperforms previous methods in terms of reconstruction and synthesis quality and expression control accuracy but the cross-reenactment identity consistency is slightly worse than some existing methods. We believe this is due to the 3DMM [Li et al., 2017] and 3DMM tracker we rely on, whose identity parameters and expression parameters are not completely decoupled. Some methods [Deng et al., 2024b,a] that are not based on 3DMM have brought some inspiration to solve this limitation, and we leave these to future work. Importantly, our model not only achieves these quantitative results, but also achieves the real-time reenactment speed, which is much faster than existing methods.

**Inference speed and efficiency.** Our method can run at 67 FPS on an A100 GPU with the naive PyTorch framework and official 3D Gaussian Splatting implementation. As shown in Tab. 3, we are the first real-time method for animatable one-shot head avatar reconstruction, which shows the application prospects and unique value of our method.

Table 3: The time of reenactment is measured in FPS. All results exclude the time for getting driving parameters that can be calculated in advance and are averaged over 100 frames.

| | StyleHeat | ROME | OTAvatar | HideNeRF | GOHA | CVTHead | GPAvatar | Real3D | P4D | P4D-v2 | Ours |
|---|---|---|---|---|---|---|---|---|---|---|---|
| Driving FPS | 19.82 | 11.21 | 0.12 | 9.73 | 6.57 | 18.09 | 16.86 | 4.55 | 9.49 | 9.62 | 67.12 |

## 4.3 Ablation Studies

**Dual-lifting.** To validate the effectiveness of our proposed dual-lifting method, we compare it against a baseline that lifts points from a single plane. This baseline requires the model to simultaneously lift points forward and backward from the image plane, sometimes creating ambiguities. The results in Tab. 4 and Fig. 4 show that dual-lifting significantly enhances reconstruction quality. Moreover, since the lifting is performed only once per identity and subsequent expression driving does not require recalculation, dual-lifting does not impact the performance during reenactment.

Table 4: Ablation results on the VFHQ [Xie et al., 2022] dataset.

| Method | Self Reenactment | | | | | | | Cross Reenactment | | |
|---|---|---|---|---|---|---|---|---|---|---|
| | PSNR↑ | SSIM↑ | LPIPS↓ | CSIM↑ | AED↓ | APD↓ | AKD↓ | CSIM↑ | AED↓ | APD↓ |
| one-plane lifting | 21.34 | 0.802 | 0.158 | 0.781 | 0.127 | 0.170 | 3.810 | 0.581 | 0.272 | 0.290 |
| w/o $F_{id}$ | 21.13 | 0.807 | 0.155 | 0.774 | 0.125 | 0.155 | 3.722 | 0.537 | 0.270 | 0.272 |
| w/o neural renderer | 20.34 | 0.789 | 0.138 | 0.788 | 0.147 | 0.202 | 4.763 | 0.623 | 0.300 | 0.353 |
| w/o $\mathcal{L}_{lifting}$ | 21.64 | 0.812 | 0.148 | 0.800 | 0.119 | 0.151 | 3.563 | 0.620 | 0.261 | 0.252 |
| Ours | 21.83 | 0.818 | 0.122 | 0.816 | 0.111 | 0.135 | 3.349 | 0.633 | 0.253 | 0.247 |

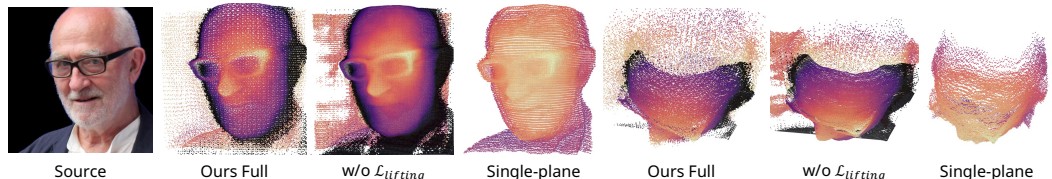

| Source | Ours Full | w/o $\mathcal{L}_{lifting}$ | Single-plane | Ours Full | w/o $\mathcal{L}_{lifting}$ | Single-plane |

Figure 5: Lifting results of an in-the-wild image, include the front view and the top view. Points are filtered by Gaussian opacity. We color two parts of the dual-lifting separately, and the black points are the image plane. It can be seen that the lifted 3D structure is relatively flat without $\mathcal{L}_{lifting}$.

**Lifting distance loss.** We evaluate the influence of the lifting distance loss $\mathcal{L}_{lifting}$ by removing it during training. Without lifting distance loss, we observed performance degradation as shown in Tab. 4 and Fig. 4. Compared with our full method, removing the point distance constraint will make it more difficult to reconstruct high-quality 3D structures, especially on facial edges.

**3D structure of dual-lifting.** We further analyze and compare the 3D structure of dual-lifting. We show the visualization of filtered lifting points in Fig. 5. It can be seen that in the case of single-plane lifting or without $\mathcal{L}_{lifting}$, the model can learn the correct 3D lifting even without any explicit 3D constraints. However, dual-lifting can produce 3D Gaussian points away from the input angle, and the 3D structure is also more reasonable rather than relatively flat.

**Global feature in expression branch.** We conduct an ablation study by removing the global identity features $F_{id}$ from the expression branch. The results in Tab. 4 and Fig. 4 indicate that removing $F_{id}$ decreases the identity similarity (CSIM) of the results and the reenactment quality. This demonstrates the importance of incorporating identity information in the expression branch.

**Neural renderer.** Due to the sparsity of our reconstructed Gaussians, we increased the output dimensions and introduced a neural renderer to refine the coarse images and features. This process is similar to the super-resolution module in EG3D [Chan et al., 2022], but our neural renderer does not increase the resolution of the results. The results in Tab. 4 and Fig. 4 show the performance of coarse results without neural rendering. It can be observed that we can obtain reasonable results even using only sparse Gaussians, but the neural renderer significantly improves detail and expression.

**Extreme inputs.** We present more qualitative results with extreme inputs in Fig. 6. For extreme expressions or common occlusions such as sunglasses, our method shows good robustness. Our model can also work well with low-quality images and challenging lighting conditions, but the details of reconstructed avatars are inevitably affected. For example, avatars reconstructed from blurred images lack details, while those from images with challenging lighting conditions have fixed lighting, such as shadows on the nose. However, these features also demonstrate that our method can faithfully restore details and handle various extreme cases.

**Resolve conflicts of dual-lifting and expression Gaussians.** Although we attempt to bring the two sets of Gaussians closer, there are inherent conflicts since one set is static and the other is dynamic. We show some results with conflicts in Fig. 7. It can be seen that the RGB values conflict when there is a significant expression difference between the dual-lifting Gaussians and the expression Gaussians, but these conflicts are well resolved after neural rendering. We believe this is because our Gaussians have 32-D features that contain more information than RGB values. The neural rendering module can act as a filter to integrate the two point sets using these features and resolving possible conflicts.

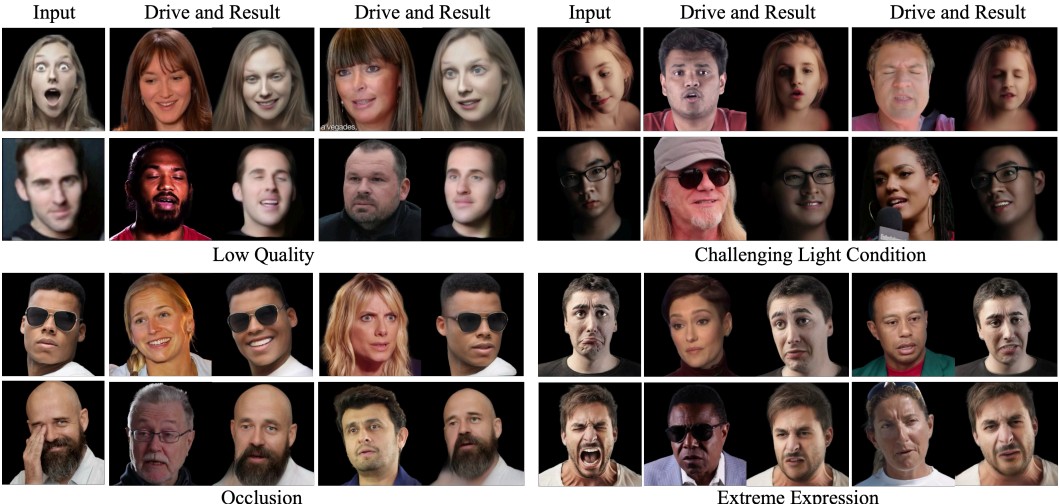

| Input | Drive and Result | Drive and Result | Input | Drive and Result | Drive and Result |

Low Quality — Challenging Light Condition

Occlusion — Extreme Expression

Figure 6: The robustness of our model. Our method can produce reasonable results for low-quality images, challenging lighting conditions, significant occlusions, and extreme expressions.

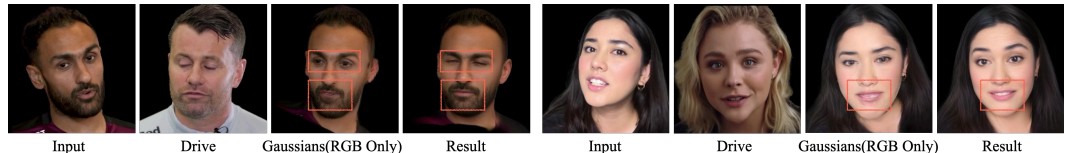

| Input | Drive | Gaussians(RGB Only) | Result | Input | Drive | Gaussians(RGB Only) | Result |

Figure 7: The case where two sets of Gaussians conflict, the conflict is resolved after neural rendering. We believe that neural rendering resolves the conflict through the 32D features carried by Gaussians.

## 5  Conclusion

In this paper, we proposed a novel framework for one-shot head avatar reconstruction and real-time reenactment. The key innovation of our method is the dual-lifting approach for one-shot 3D Gaussian reconstruction, which estimates the Gaussian parameters in a single forward pass. We also propose a 3DMM-based expression control method and a loss function that uses 3DMM priors to constrain the lifting process. Our experiments demonstrate that our method outperforms state-of-the-art baselines in both the quality of head avatar reconstruction and reenactment accuracy, with significant improvements in rendering speed. We believe our method has a wide range of potential applications due to its strong generalization capabilities and real-time rendering speed.

**Broader impacts.** Although our method has great potential in various applications, it also poses the risk of misuse, such as generating fake videos and spreading false information. We strongly oppose such misuse and have proposed several measures to prevent it, as detailed in Sec. E. With proper and responsible use, we believe our method can offer significant benefits in a wide range of applications such as video conferencing and entertainment industries.

**Limitations and future work.** Despite its strengths, our method has certain limitations. For example our model may generate less detail for unseen areas, and our 3DMM-based expression branch cannot control the areas not modeled by 3DMM, such as hair and tongue. These limitations highlight the possible improvements in future work to increase the performance and practicality of our method. In Sec. F, we provide a more detailed discussion of our limitations and future work.

## Acknowledgements

This work was partially supported by JST Moonshot R&D Grant Number JPMJPS2011, CREST Grant Number JPMJCR2015 and Basic Research Grant (Super AI) of Institute for AI and Beyond of the University of Tokyo. In addition, this work was also partially supported by JST SPRING, Grant Number JPMJSP2108.

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

# A    Reproducibility

## A.1    More Implementation Details

Specifically, we use DINOv2 Base as our feature extractor, which takes $3 \times 518 \times 518$ images as input, and encodes $296 \times 296$ local feature maps and 768-dimensional global features. We then obtain the Gaussian parameters of each pixel through 4 groups of ResBlocks He et al. [2016] without down-sampling. The dimension of Gaussian parameters is 41 dimensions, including 32 dimensions of color information, 1 dimension of opacity information, 3 dimensions of scale information, 4 dimensions of rotation information, and 1 dimension of lifting distance information. Since FLAME [Li et al., 2017] contains 5023 points, we assign a 256-dimensional feature to each point, so the total point feature size is $5023 \times 256$. We concatenate these features with global features to predict expression Gaussian parameters using an MLP with 1024 input dimensions. This MLP contains 6 layers, and since it does not include lifting distance, the output is 40 dimensions. Our neural renderer employs StyleUNet [Wang et al., 2021b] to map images from $32 \times 512 \times 512$ to $3 \times 512 \times 512$ dimensions. We also provide the code for the model in the supplementary material for reference.

## A.2    More Data Processing Details

We use 15,204 video clips from the VFHQ dataset [Xie et al., 2022] for training and 100 videos for testing, following the original VFHQ split. For training videos, we uniformly sample frames based on the video's length: 25 frames if the video is less than 2 seconds, 50 frames if the video is 2 to 3 seconds, and 75 frames if the video is longer than 3 seconds. For testing videos, we uniformly sample 50 frames from each clip, resulting in a total of frames for training and 5,000 frames for testing. For the HDTF dataset, we use the test split from OTAvatar [Ma et al., 2023], which includes 19 videos. We uniformly sample 100 frames from each video, creating a test set with 1,900 frames.

For all these frames, we remove the background and resize them to $512 \times 512$ pixels. We extract and refine the 3DMM parameters for each frame following [Chu et al., 2024]. Although the labels generated by this automatic annotation method are somewhat noisy and imperfect, this approach allows us to build a large dataset, effectively mitigating the impact of data inaccuracies.

## A.3    More Evaluation Details

We conduct comparisons with several state-of-the-art methods, including ROME [Khakhulin et al., 2022], StyleHeat [Yin et al., 2022], OTAvatar [Ma et al., 2023], HideNeRF [Li et al., 2023b], GOHA [Li et al., 2023a], CVTHead [Ma et al., 2024a], GPAvatar [Chu et al., 2024], Real3DPortrait [Ye et al., 2024], Portrait4D [Deng et al., 2024b], and Portrait4D-v2 [Deng et al., 2024a]. For each method, we use the official data pre-processing script to process its input frame and driver frame, and use the official implementation to obtain the result frame. To ensure a fair comparison, we realign the results from all methods, as some methods crop and center the face region while others do not. Specifically, we detect landmarks and crop the head region at the same size for all driving images and results, and then resize the results to 512×512 for evaluation.

It is worth noting that although Portrait4D and Portrait4D-v2 achieve the same functionality and get really good results, their core contributions are orthogonal to our work. They introduce a new data generation method and a new learning paradigm, which means our method can also benefit from their advancements. We leave the integration of these parallel works to future research.

# B    Preliminaries of 3DMM

We utilize a widely-used 3D morphable model (3DMM): the FLAME [Li et al., 2017] model which renowned for its geometric accuracy and versatility. This model is popular in applications such as facial animation, avatar creation, and facial recognition due to its realistic rendering capabilities and flexibility. We use it to work as our expression driven signal and geometry prior. The FLAME model represents the head shape as follows:

$$TP(\hat{\beta}, \hat{\theta}, \hat{\psi}) = \bar{T} + BS(\hat{\beta}; S) + BP(\hat{\theta}; P) + BE(\hat{\psi}; E), \tag{5}$$

where $\bar{T}$ is the template head avatar mesh, $BS(\hat{\beta}; S)$ is the shape blend-shape function to account for identity-related shape variation, $BP(\hat{\theta}; P)$ is a jaw and neck pose part to correct pose deformations

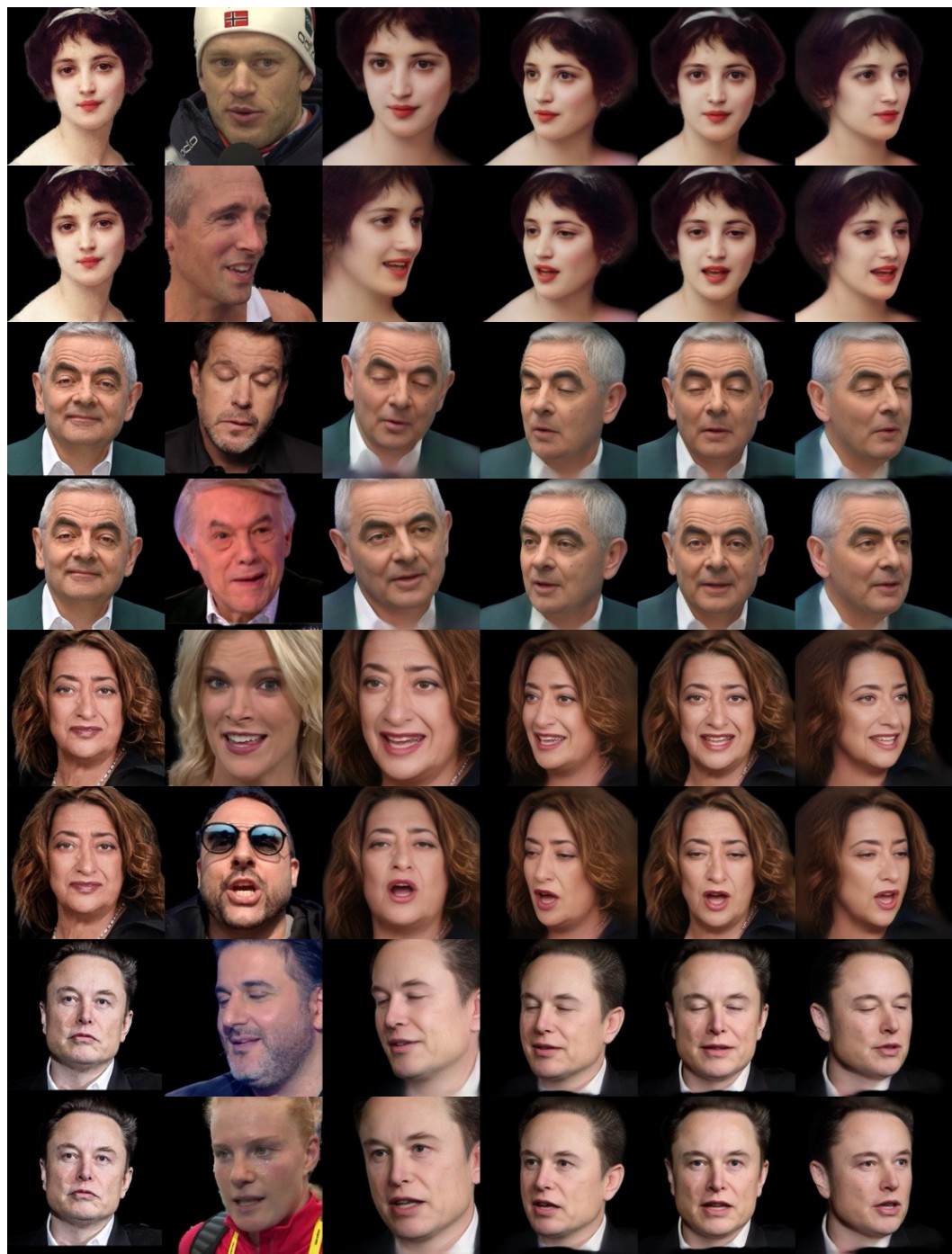

Figure 8: Reenactment and multi-view results of our method on in-the-wild images. From left to right: input image, driving image, driving and novel view results.

that cannot be explained solely by linear blend skinning, and expression blend-shapes $BE(\hat{\psi}; E)$ is used to capture facial expressions such as closing eyes or smiling.

## C   Per-part Rendering and 3D Lifting of Our Method

We present the results of rendering the dual-lifting Gaussians from the reconstruction branch and the Gaussian from the expression branch separately. As Fig. 9 shows, the dual-lifting Gaussians

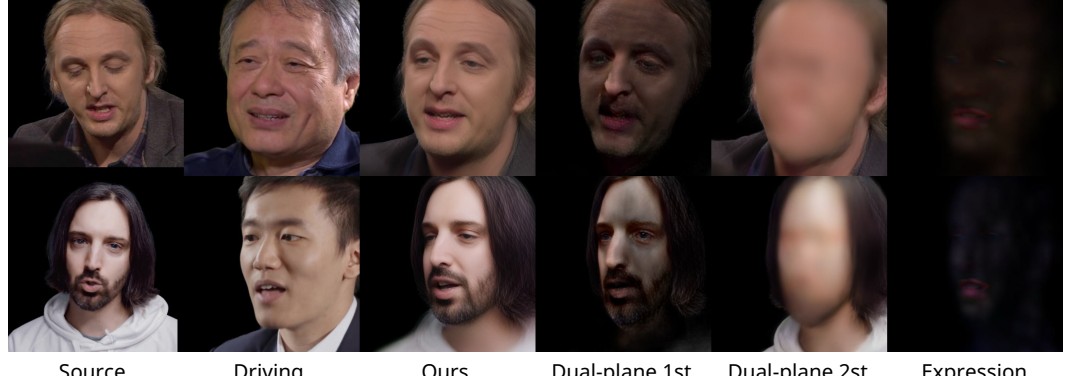

| Source | Driving | Ours | Dual-plane 1st | Dual-plane 2st | Expression |

Figure 9: Per-part rendering of the dual-lifting and expression Gaussians. We can see that the dual-lifting Gaussians reconstruct the head's base structure and facial details respectively. It is worth noting that our Gaussians are not purely RGB Gaussians. Instead, our Gaussians include 32-D features (as described in Sec. 3.3). We visualize the first 3 dimensions of these features (i.e., the RGB values of the Gaussians) here without the neural rendering module. So this visualization is intended to intuitively display the functionality of each part and the importance of each branch should not be judged based on RGB values alone.

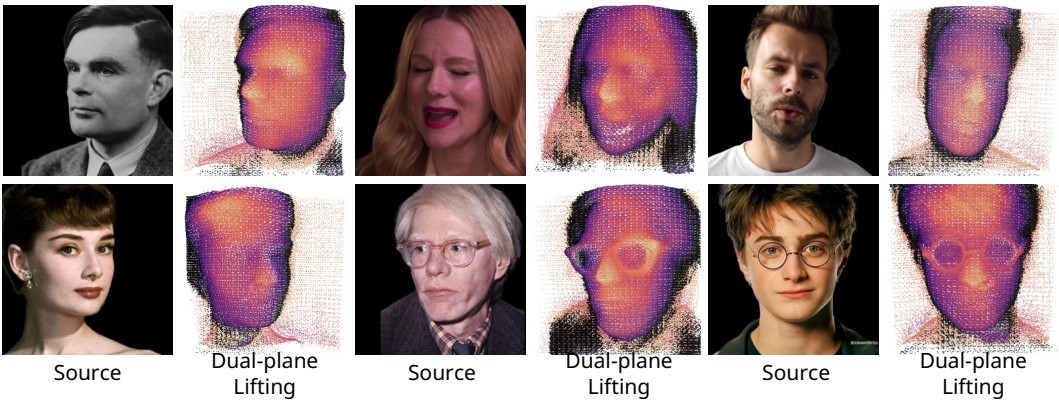

| Source | Dual-plane Lifting | Source | Dual-plane Lifting | Source | Dual-plane Lifting |

Figure 10: Dual-lifting results of in-the-wild images. We can see that the dual-lifting point cloud has rich details, including glasses and hair. We color the two parts of the dual-lifting separately, and the black points are the image plane.

reconstruct the head's base structure and facial details respectively, which is in line with our expectations. We also show more lifted points in Fig. 10, we can see that the dual-lifting point cloud has rich details, including glasses and hair. Additionally, we provide some lifting point cloud files in supplementary materials.

# D   More Qualitative Results

We show more self-identity qualitative comparisons with baseline methods in Fig. 11, and cross-identity qualitative comparisons in Fig. 13. Here we show the results of all baseline methods on the VFHQ [Xie et al., 2022] dataset and HDTF [Zhang et al., 2021] dataset.

We also show more results of our method and baseline methods for self and cross-identity reenactment. In Fig. 12, we not only show the reenactment results but also the multi-view results of our method. In Fig. 16, we show more comparisons and consecutive frames and highlight the regions of interest. We also show more in-the-wild results of our method in Fig. 8, 14 and 15. It can be seen that our method maintains good identity consistency and 3D consistency when the viewing angle changes.

Additionally, we provide a supplementary video to demonstrate video driving results. Although no special processing is performed, our method has timing-stable results on video generation.

Source Driving StyleHeat ROME OTAvatar HideNeRF GOHA CVTHead GPAvatar Real3D P4D P4D-v2 Ours

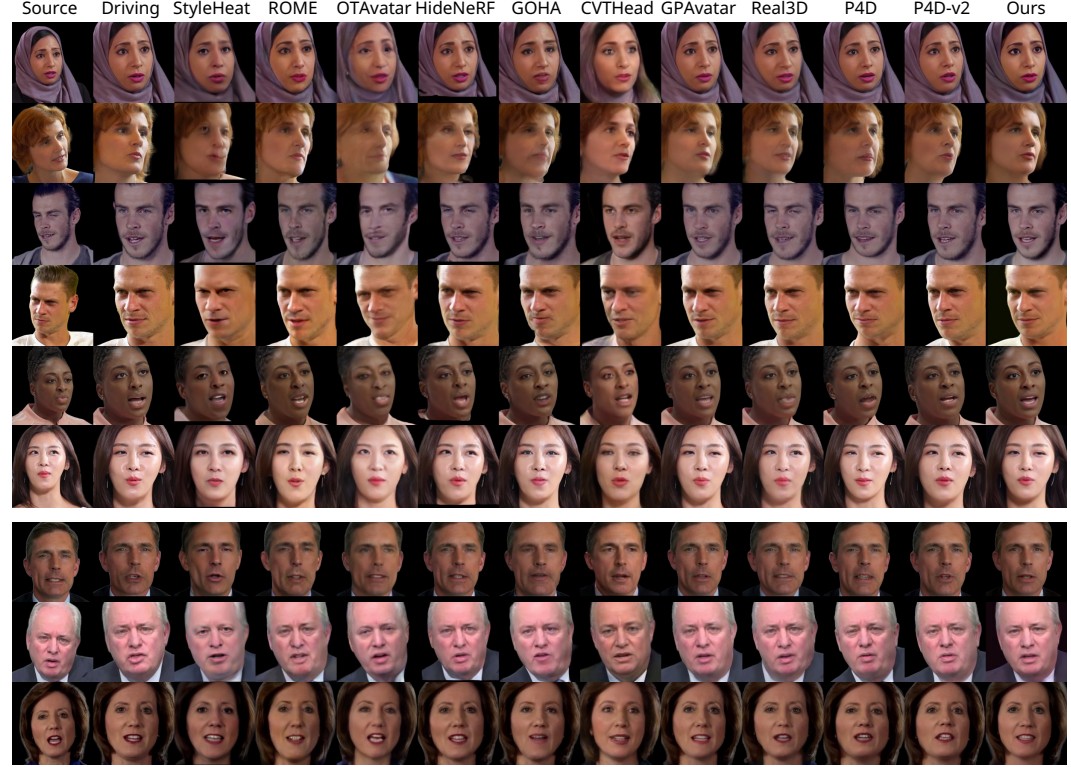

Figure 11: Self-identity reenactment results on VFHQ [Xie et al., 2022] and HDTF [Zhang et al., 2021] datasets. The top six rows are from VFHQ and the bottom three rows are from HDTF.

Source Driving Reenactment Novel Views    Source Driving Reenactment Novel Views

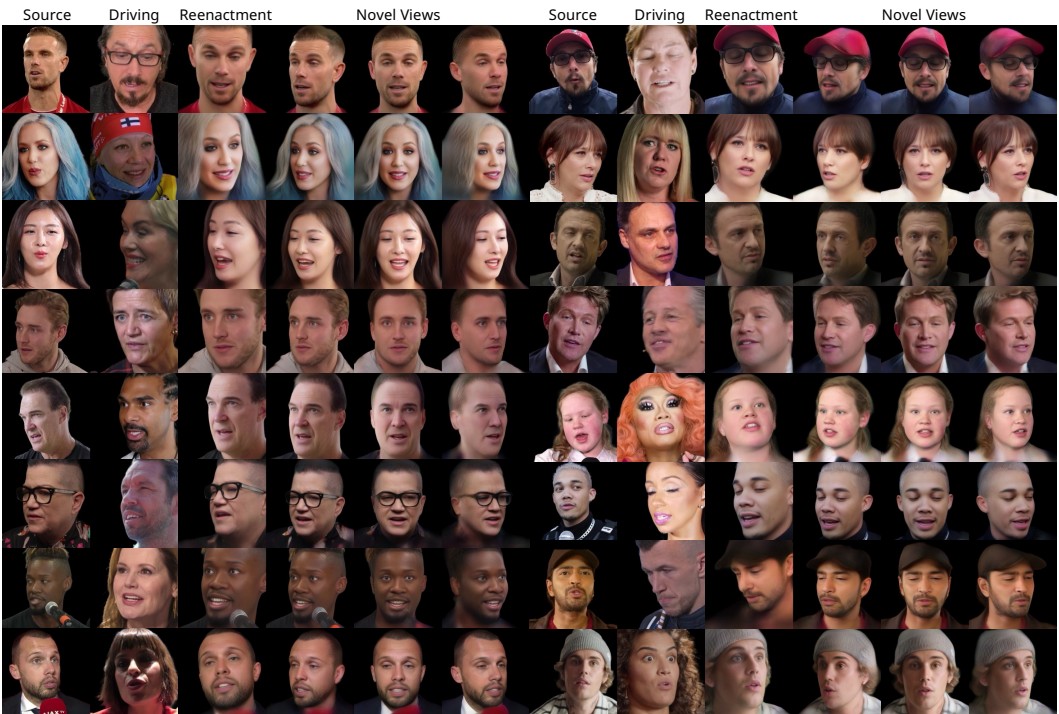

Figure 12: Reenactment and multi-view results of our method on the VFHQ [Xie et al., 2022] dataset. Our method can maintain consistency across multiple views.

Source  Driving StyleHeat ROME  OTAvatar HideNeRF GOHA CVTHead GPAvatar Real3D  P4D  P4D-v2  Ours

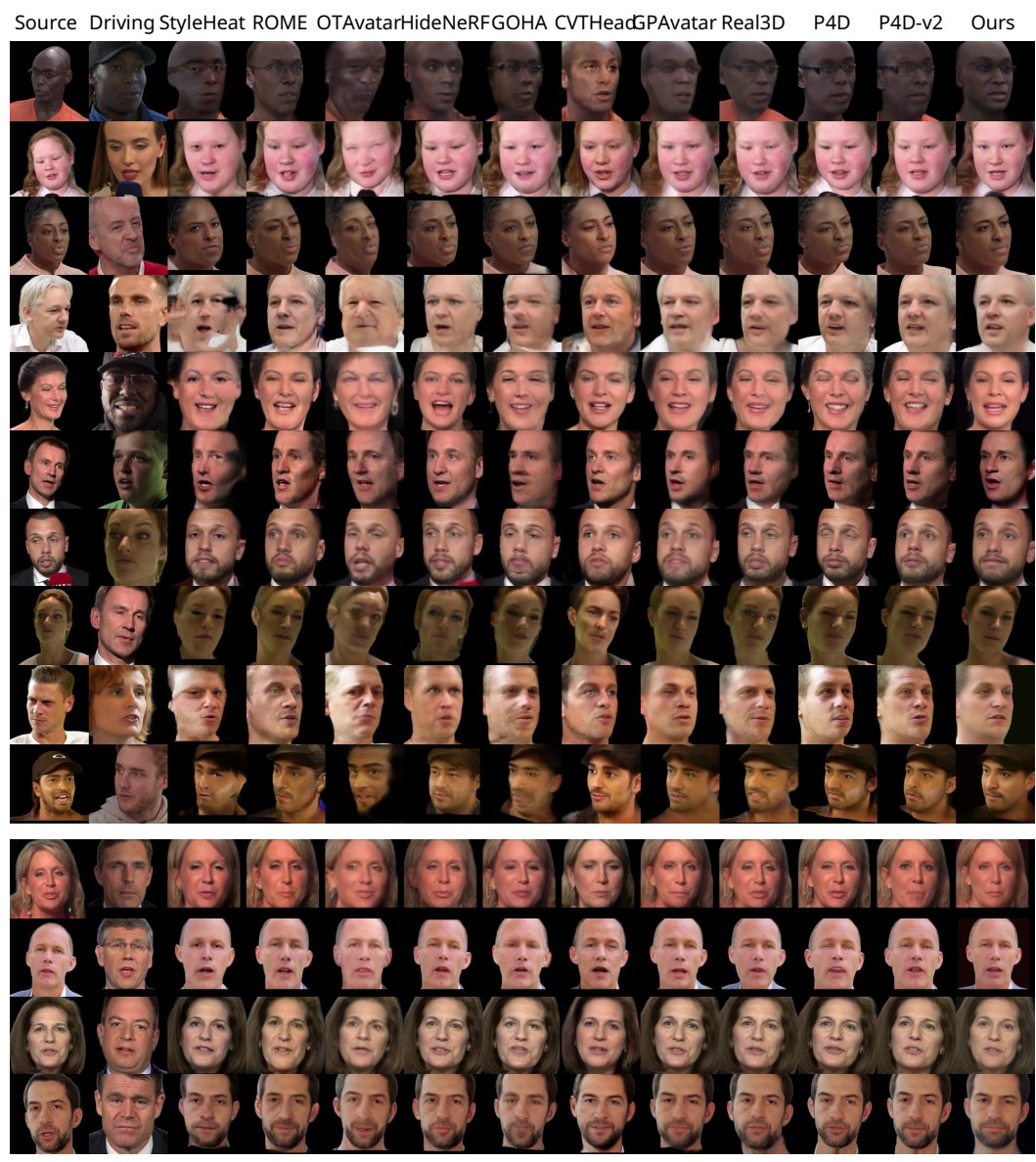

Figure 13: Cross-identity reenactment results on VFHQ [Xie et al., 2022] and HDTF [Zhang et al., 2021] datasets. The top ten rows are from VFHQ and the bottom four rows are from HDTF.

# E   More In-Depth Ethical Discussion

Our framework offers many applications but also presents ethical risks, such as the potential creation of fake videos ("deepfakes"), violations of privacy, and the dissemination of false information. We do not advocate such misuse and have proposed several measures to prevent these risks:

**Watermarking technologies.** To ensure transparency and prevent misuse, we plan to employ watermarking techniques in code that will be released. Visible watermarks enable viewers to immediately recognize content as AI-generated, helping them distinguish potential misuse. In addition to visible watermarks, we plan to embed invisible watermarks [Tancik et al., 2020], which are difficult to remove. These invisible marks help track and identify the source of videos, even if they are re-edited. This tracking capability encourages producers to consider the ethical implications and potential risks of their creations by storing information about the video producer.

**Strict licenses.** We will release our code and model under a strict license. The license will prohibit the synthesis of real individuals without explicit consent for commercial use. This ensures that our

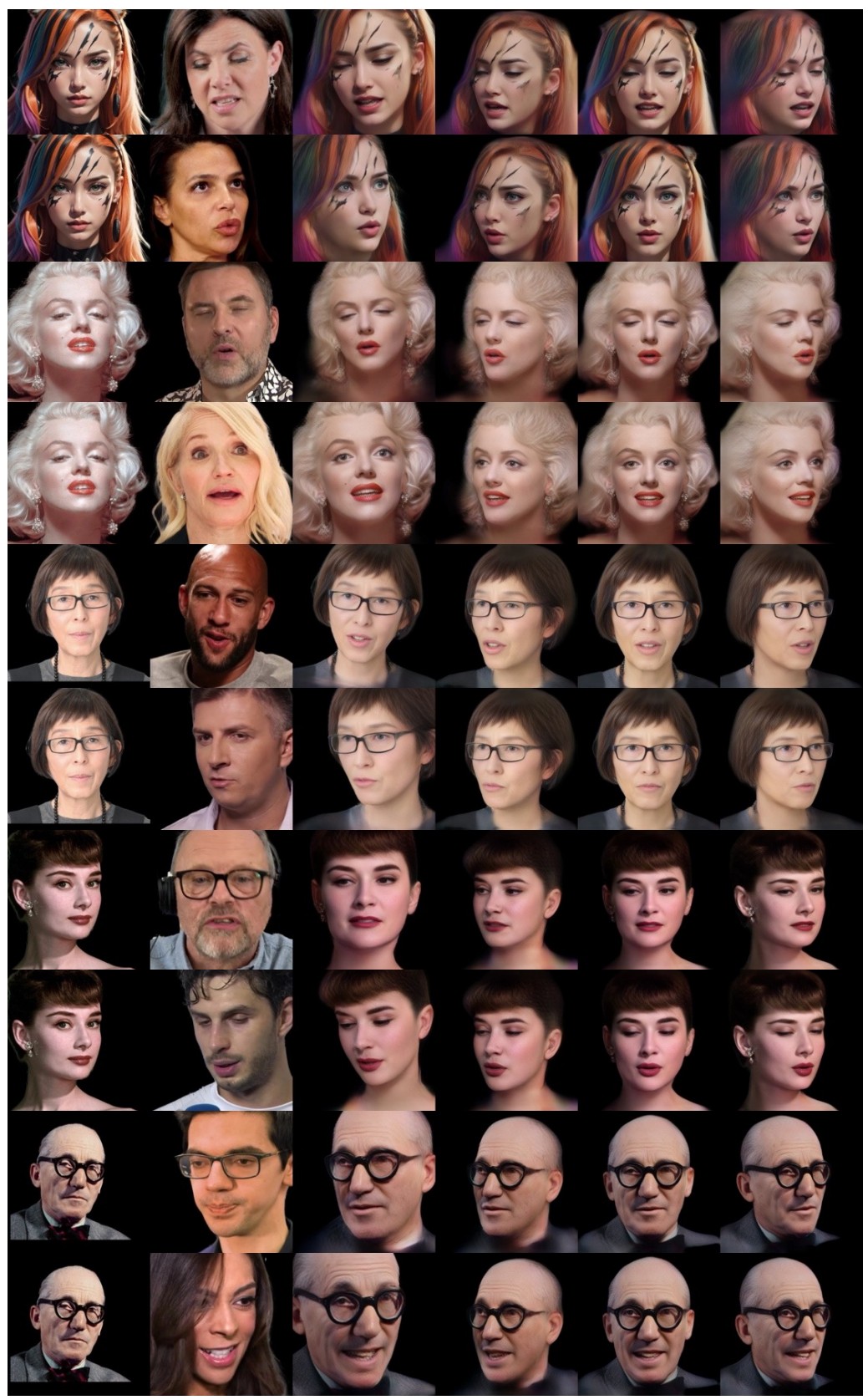

Figure 14: Reenactment and multi-view results of our method on in-the-wild images. From left to right: input image, driving image, driving and novel view results.

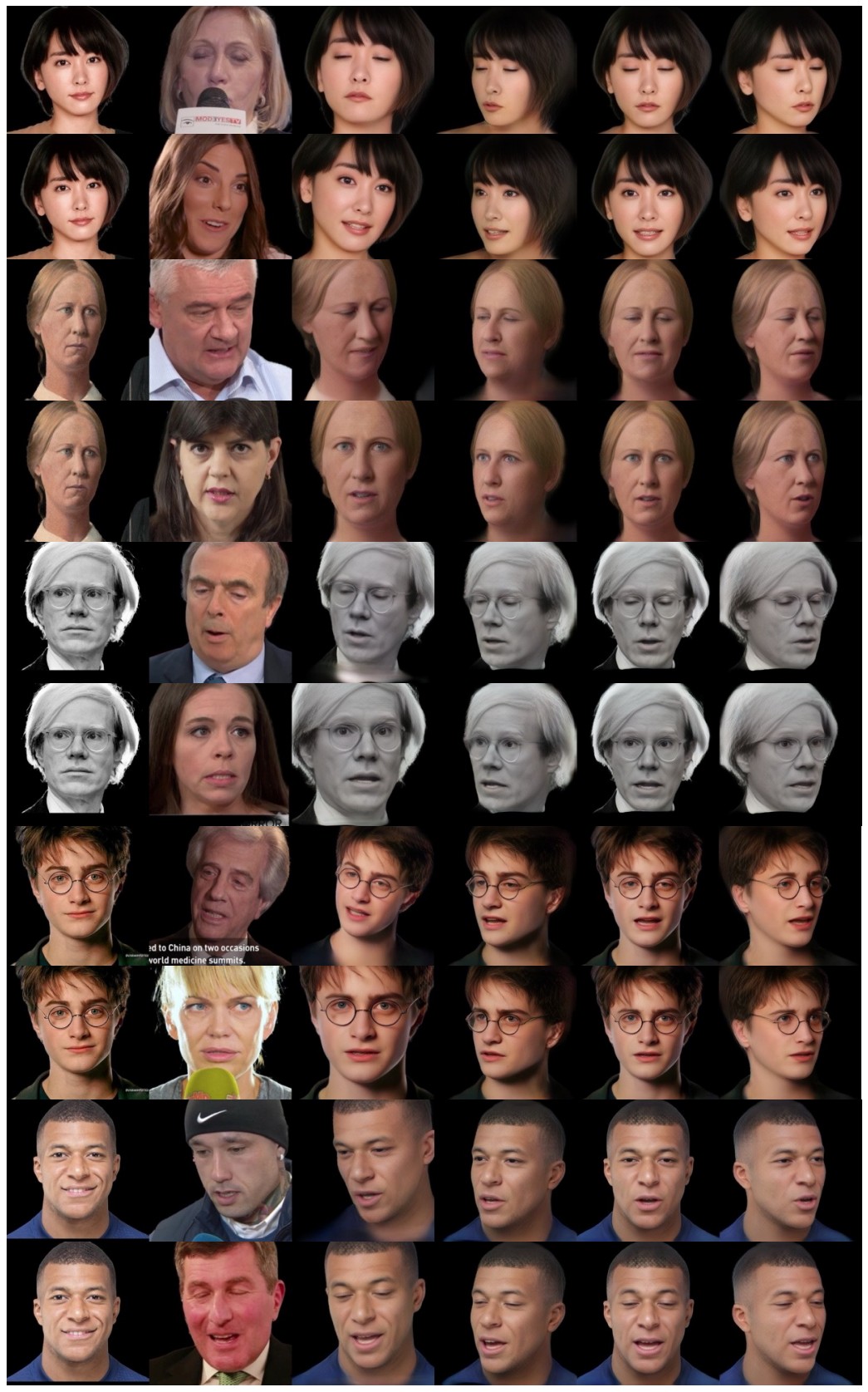

Figure 15: Reenactment and multi-view results of our method on in-the-wild images. From left to right: input image, driving image, driving and novel view results.

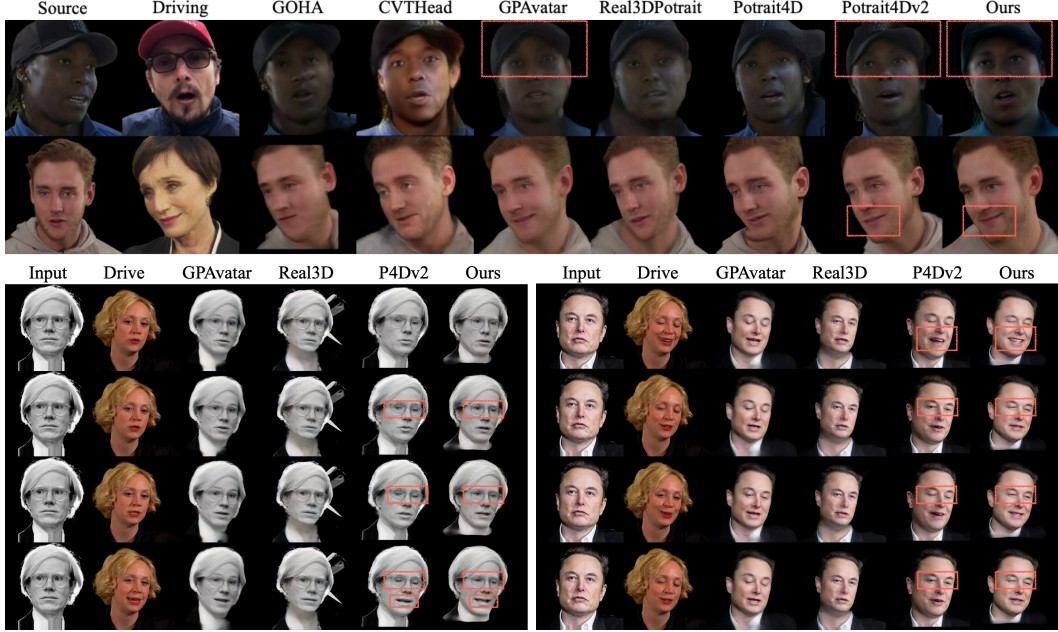

Figure 16: Qualitative results and video continuous frame results with highlighted attention regions. We selected competitive methods to show continuous frames. Better to view it zoomed in.

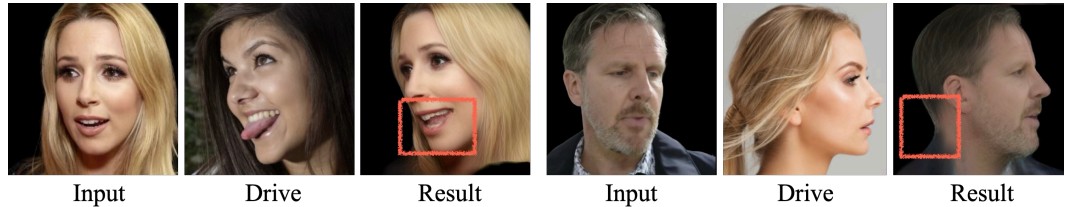

Figure 17: Our model has some limitations. For example, the tongue is not modeled and the unseen regions of the input have less details. Better to view it zoomed in.

technology is used ethically and prevents it from violating the consent of the individual represented by the avatar. Illegal misuse can be traced through the watermark system.

In summary, we will implement robust safeguards to prevent the misuse of our head avatar reconstruction system. We urge video creators to be mindful of the ethical responsibilities and potential risks when using talking face generation technologies. With careful and responsible use, our method can provide substantial benefits across various real-world applications.

## F   Limitations and Future Work

Although our method achieves high-quality synthesis results compared to previous approaches, there are still some limitations. When rendering synthesized results from novel views, unseen areas in the original source image often lack detail and may produce results with statistically average expectations. For example, generating the other half of the face from a side view input or generating an open mouth from an input image with a closed mouth. Additionally, our expression branch is based on 3DMM and learned from VFHQ video data. This branch may not capture extreme facial movements or parts not modeled by 3DMM, such as one eye being open and the other being closed, the tongue, and hair. We show the qualitative results of these limitations in Fig. 17. Future work may involve learning expression embeddings [Deng et al., 2024b] directly from images, alleviating data requirements and tracking accuracy needs through data generation [Deng et al., 2024a], gathering more expressive data to improve expression imitation. Extending our approach to handle full-body avatar synthesis is also a promising direction for future research.

