# OpenReview forum: "Generalizable and Animatable Gaussian Head Avatar"
_NeurIPS.cc/2024/Conference — NeurIPS 2024 poster_

### Official Review · Reviewer_d3t7 · 2024-07-08

**Soundness:** 3
**Presentation:** 2
**Contribution:** 2
**Rating:** 4
**Confidence:** 4

**Summary:**

This paper presents a method for animatable facial avatar reconstruction from a single RGB image, and the reconstructed avatar is based on 3D Gaussian splatting (3DGS) to support real-time rendering. To this end, the proposed method generates pixel-aligned Gaussian point clouds to reconstruct the identity, and use 3DMM to construct an additional set of Gaussian points to control the facial expression. The network is trained on a large scale corpus of human talking videos. After training, the network can produce reenactment results at real-time framerate for any unseen identities. Experiments show that the proposed method is able to produce plausible reenactment results.

**Strengths:**

* This paper proposes the first generalizable 3D Gaussian-based avatar modeling method (if not taking ECCV 2024 papers into account).

* The authors compare the proposed method with most recent state-of-the-art baselines (such as Protrait-v2). The comparison is comprehensive and convincing.

* Results in Table 1 and Table 2 show that the proposed method consistently outperforms existing approaches under differently metrics and different settings.

**Weaknesses:**

* My major concern is about the method design. Despite the plausible results, the proposed method is confusing and doesn't make sense to me. If I understand correctly, the reconstruction branch produces Gaussian points that are aligned with the input image, and these points keep static and unchanged when performing reenactment. Instead, the expressions are modeled through the additional Gaussian points that are attached to the 3DMM surface (Line 187-188), which means these expression Gaussian points should play a major role in expression control. However, according to Figure 7, the Gaussian points produced by the reconstruction branch actually play a major role in facial rendering, while the "expression Gaussians" seem to be much less important. That doesn't make sense to me: how could the static points produced by the reconstruct branch model the dynamic expressions? When overlapping the reconstruction Gaussian points with the expression ones, how does the proposed method resolve the potential conflicts of these two point sets, such as different lip positions?


* The so-called dual-lifting technique looks straight-forward and trivial to me. According to Sec.3.1, this module produces two pixel-aligned Gaussian point sets, one for the visible surface and the other for the invisible surface. However, using two set of points (visible+invisible, or front+back) to model a complete shape is not a new idea; it has been used in 3D human reconstruction like [a] many years ago. Although such an idea has not been proposed in the form of 3DGS, I don't think it is novel enough to be claimed as a technical contribution.

* It is unclear whether the baseline methods are trained on the same dataset as the proposed method or not. In Line 244 of the main paper, the authors mention that they "use the official implementation", but it is still unclear whether the authors use the pretrained network weights or retrain these baselines using the same dataset.

[a] Gabeur et al. Moulding Humans: Non-parametric 3D Human Shape Estimation from Single Images. ICCV 2019.

**Questions:**

One minor suggestion: it would be better if the authors could provide some qualitative comparison against state-of-the-art methods in the form of dynamic reenactment videos and highlight the difference/advantages of the results.

**Limitations:**

The authors have discussed the limitation and potential social impact in Sec.5 of the main paper and Sec.E & F of the supplemental document.

---

> ### Author Rebuttal · Authors · 2024-08-07
>
> Thank you for your review and helpful comments, which triggered deeper thinking about our approach. We would like to address your concerns in the following sections:
>
> **How could the static points produced by the reconstruction branch model the dynamic expressions?**
> * Your understanding is mostly correct. In our method, the reconstruction branch generates static Gaussian points, while the expression branch generates dynamic Gaussian points.
> * However, it's important to note that our Gaussians are not purely RGB or spherical harmonics Gaussians. Instead, our Gaussians include 32-D features (as described in Sec. 3.3). In Fig. 7 of the paper, we visualize the first 3 dimensions of these features (i.e., the RGB values of the Gaussians) without the neural rendering module. This visualization is intended to intuitively display the functionality of each part and the importance of each branch should not be judged based on RGB values ​​alone. Their importance is determined by the entire 32-D features and the neural renderer module. In fact, the expressions are modeled solely by the expression branch, without expression Gaussians, the lifted Gaussians from the reconstruction branch are static.
>
> **How does the proposed method resolve the potential conflicts of these two point sets?**
> * Thank you for pointing out some missing parts of the paper. We will include this discussion on how to resolve conflicts in the revised version.
> * Although we attempt to bring the two point sets closer together, there are inherent conflicts since one set is static and the other is dynamic. We address these conflicts through the neural rendering module. Our Gaussian points have 32-D features, which contain more information than just RGB values. Thus, the neural rendering module can leverage these features to integrate the two point sets. We show some results with conflicts in Fig. 4 on the new supplementary page. It can be seen that when there are significant expression differences, the RGB values of the Gaussian points conflict, but these conflicts are well-resolved after neural rendering. We believe this demonstrates that the neural rendering module acts as a filter, using features from the expression Gaussian in areas where conflicts may arise.
>
> **About the novelty of the dual-lifting method.**
> * We believe our method is novel. The work of [Gabeur et al.] uses two sets of points to model a static human body and then obtains the body mesh. Our method uses two sets of lifted 3D Gaussians to model the human head, which is the first one-shot 3DGS method for head avatars. Additionally, we incorporate expression Gaussians, enabling dynamic expression modeling, which further develops this method. Our method also achieves real-time re-driving, which is a first in one-shot head avatars and is crucial for practical applications. Finally, our work provides a comprehensive evaluation and comparison for one-shot dynamic head avatar reconstruction, which builds a solid baseline for future research.
> For missing citations and discussion of related papers, we will include and discuss them in the revised version.
>
> **The baseline methods and evaluation method.**
> * We used pre-trained network weights and the official code implementations for evaluation. Since each method has different data requirements, it is challenging to train every method on a unified dataset. For instance, Portrait4D uses single-view head images, Portrait4DV2 uses generated multi-view data, GOHA needs video and single-view head images, Real3DPortrait needs EG3D-generated multi-view head images and pre-train image-to-plane model, ROME did not release their training code.
> * Since these methods emphasize their generalization ability, using pre-trained weights for evaluation is acceptable and all compared baseline works also follow the same way for their evaluation.
> * Additionally, the HDTF and VFHQ test sets are commonly used benchmark datasets. OTAvatar, GOHA, and GPAvatar have reported results on the HDTF test set, while GPAvatar, Portrait4D, and Portrait4DV2 have reported results on the VFHQ test set, therefore we chose to test on these two datasets.
>
> **Side-by-side video comparison with baselines.**
> * Due to rebuttal restrictions, we are unable to provide additional videos for rebuttal. Instead, we have included some consecutive frames in the new supplementary page and highlighted the areas of interest. We will add side-by-side video comparisons in our future open-source code and demo website.

---

### Official Review · Reviewer_65B2 · 2024-07-08

**Soundness:** 3
**Presentation:** 3
**Contribution:** 3
**Rating:** 5
**Confidence:** 3

**Summary:**

The paper proposes a method to achieve one-shot head avatar animation with 3D Gaussian Splatting (3DGS). With 3DGS, the papers show high-fidelity animation with fast inference speed. To solve one-shot 3DGS reconstruction, the paper proposes a dual-lifting method with 3DMM regularization. Experiments justify the model designs and show the proposed approach achieves SOTA performance.

**Strengths:**

1. The paper proposes a technically sound approach for solving the one-shot animatable head avatar problem using 3D Gaussian Splatting (3DGS), fully utilizing the pre-trained 2D network to obtain cues for lifting.
2. The quantitative results are promising, and the qualitative results are sufficient to demonstrate the robustness of the approach.

**Weaknesses:**

1. The visual results of the method seem to be overly smoothed to some extent, making the results less realistic compared to some baselines (e.g., Portrait4Dv2). The authors may need to provide additional explanation for this and discuss why the method achieves better qualitative results than other methods.
2. The approach is highly dependent on 3DMM but lacks a discussion about its impact. What if the estimation fails? How does the estimation accuracy affect the final performance (both training and inference)? If using other designs without 3DMM, what would be the method's performance?
3. There should be some failure cases to show the limitations of the work.

**Questions:**

1. The current ablation study may not be sufficient to validate the dual-plane lifting. Is it possible to conduct a study by removing the reconstruction branch to better demonstrate the importance of dual-plane lifting?

**Limitations:**

Yes, the authors discuss their "Limitations" and provide an "In-Depth Ethical Discussion" proposing several measures to prevent these ethical risks.

---

> ### Author Rebuttal · Authors · 2024-08-07
>
> Thank you for your positive review and helpful comments. We would like to address your concerns in the following sections:
>
> **Is it possible to conduct a study by removing the reconstruction branch to better demonstrate the importance of dual-plane lifting?**
>
> * Removing the reconstruction branch is possible, but it would result in a lack of detail. Our method integrates global identity features when predicting the expression Gaussians, which provides some identity information to the expression Gaussians. However, the 1D global features are insufficient to reconstruct detailed identity features. We provide qualitative results of the reconstruction branch in Fig. 2 on the new supplementary page and quantitative results here. The qualitative results show a severe lack of details on identity, demonstrating the necessity of the reconstruction branch, and the quantitative results also support this conclusion (the CSIM metric).
> * | Method | PSNR↑ | SSIM↑ | LPIPS↓ | CSIM↑ | AED↓ | APD↓ | AKD↓ | CSIM↑ | AED↓ | APD↓ |
> | ---- | ---- | ---- | ---- | ---- | ---- | ---- | ---- | ---- | ---- | ---- |
> | w/o Recons | 18.006 | 0.756 | 0.261 | 0.454 | 0.203 | 0.223 | 5.324 | 0.230 | 0.246 | 0.279 |
> | Ours (full) | **21.83** | **0.818** | **0.122** | **0.816** | **0.111** | **0.135** | **3.349** | **0.633** | **0.253** | **0.247** |
>
> **The visual results of the method seem to be overly smoothed, making the results less realistic compared to some baselines. The authors may need to provide additional explanation for this and discuss why the method achieves better qualitative results than other methods.**
> * Currently used quantitative metrics focus more on the correctness of results rather than their realism. For example, AED measures whether subtle expressions are correctly modeled, and PSNR and SSIM measure if identity details are authentically reproduced. Sometimes visually realistic results may be incorrect. To verify the realism of our results, we have provided our quantitative evaluation with FID scores (self-reenactment on VFHQ), which is a commonly used metric for realism. Our model demonstrates competitive performance compared to state-of-the-art methods.
> * | Method | StyleHeat | ROME | OTAvatar | HideNeRF | GOHA | CVTHead | GPAvatar | Real3D | P4D | P4D-v2 | Ours |
> | ---- | ---- | ---- | ---- | ---- | ---- | ---- | ---- | ---- | ---- | ---- | ---- |
> | **FID↓** | 72.138 | 49.516 | 70.692 | 51.930 | 39.638 |109.054 | 37.610 | 38.999 | 46.965 | 30.573 | **28.938** |
>
> * Our method tends to over-smooth the hair regions. We believe this because our method does not include control of these regions. During training, the hair and upper body will always look like the input image instead of the target image, and this inconsistency leads to over-smoothing. This problem is also observed in other baseline methods since all the baseline methods do not support hair control. Among them, GOHA, Real3DPotrait, and Potrait4D use GAN loss to enhance the realism, but lead to inaccurate expression control and false illusion of details, while Portrait4DV2 improves hair realism by using data generation to obtain correct ground truth of uncontrolled regions. But it's worth noting that the primary contribution of Portrait4DV2 is a new learning paradigm, which is orthogonal to our main contribution. Our method can also benefit from the data generation method of Portrait4DV2.
>
> **The approach is highly dependent on 3DMM but lacks a discussion about its impact.**
> * Thank you for pointing out some missing parts of the paper. We will include this discussion on 3DMM in the revised version.
> * 3DMM estimation is important in our method. We use the 3DMM estimation methods provided by GPAvatar (based on EMOCA and MICA) to process our training and inference data. While these methods introduce some errors during our training process, their robustness ensures that our model can still be effectively trained. In the future, our method will also benefit from advancements in 3DMM estimation; the more accurate the estimation, the better our model is expected to perform.
> * During inference, if the 3DMM model fails to accurately predict the target expression from the target image, our model will also reproduce these inaccuracies in the driving result. For example, some subtle expressions are not captured by 3DMM (subtle frowns), or the expression is incorrect due to the non-decoupling of identity and expression. Mitigating this issue depends on further developments in the 3DMM estimation field.
> * Control without the 3DMM is possible but would require significant modifications to the method. We hope to leave this for future work.
>
> **There should be some failure cases to show the limitations of the work.**
> * We show some limitations in Fig. 3 on the new supplementary page. This includes the results of the tongue that are not modeled by the 3DMM, and the model's lack of details in invisible areas in the input image. These examples will also be integrated into future revisions.

---

> > ### Comment · Reviewer_65B2 · 2024-08-13
> >
> > Thank you for the rebuttal. After carefully reviewing the comments from the other reviewers and the authors' responses, I have decided to maintain my rating of borderline accept.

---

> > > ### Author Response · Authors · 2024-08-13
> > >
> > > Thank you for your thoughtful feedback and for taking the time to review our rebuttal and engage with us. Your comments are important for us. We are pleased we could address your concerns.
> > >
> > > We are happy to address any further concerns.

---

### Official Review · Reviewer_3QbB · 2024-07-11

**Soundness:** 3
**Presentation:** 3
**Contribution:** 3
**Rating:** 6
**Confidence:** 4

**Summary:**

The paper introduces a novel framework, GAGA, for one-shot animatable head avatar reconstruction from a single image. The key innovation is the dual-lifting method, which generates high-fidelity 3D Gaussians that capture identity and facial details by predicting lifting distances from the image plane to 3D space. This method leverages global image features and a 3D morphable model (3DMM) to ensure accurate expression control and real-time reenactment speeds. The model can generalize to unseen identities without specific optimizations. Experimental results demonstrate that GAGA outperforms previous methods in terms of reconstruction quality and expression accuracy. The main contributions include the introduction of the dual-lifting method, the use of 3DMM priors to constrain the lifting process, and the combination of 3DMM priors with 3D Gaussians for efficient expression transfer, allowing high-quality real-time rendering.

**Strengths:**

- The proposed dual-lifting of approach is interesting and enables the hallucination of unseen areas and resolves ambiguity of front/back 3D Gaussians.
- The combination of expression Gaussians and dual-lifted Gaussians allows flexible expression control.
- The method outperforms various baselines in a wide range of metrics. The rendering speed also significantly outperforms prior methods.
- The paper is clearly written and easy to follow.

**Weaknesses:**

- The Gaussians from the expression branch only use the vertices of the 3DMM model, which can be inaccurate and unable to model fine facial features.
- It would be nice to show more side-by-side video comparison with baselines. Many baselines look similar qualitatively and it’s hard to tell the improvement without highlighting the difference.

**Questions:**

- Why not also predict position offsets for the expression Gaussians? To keep expression structures, regularization can be used.
- Would the two sets of Gaussians (expression & dual-lifted) contract each other during expression?
- Why not bind the lifted Gaussians to the 3DMM vertices and animate together?

**Limitations:**

The paper acknowledges several limitations. First, the model may generate less detailed reconstructions for unseen areas of the face, such as the back of the head or the interior of the mouth, which are not visible in the input image. Second, the 3DMM-based expression branch cannot control parts of the head that are not modeled by 3DMM, such as hair and the tongue.

---

> ### Author Rebuttal · Authors · 2024-08-07
>
> Thank you for your positive review and insightful comments. We are happy to address your questions in the following:
>
> **Why not also predict position offsets for the expression Gaussians?**
> * Our method emphasizes the efficiency of inference rendering. Predicting expression Gaussians offset for each frame will impact the inference efficiency. Our expression Gaussians are controlled by 3DMM vertices now, which can be performed asynchronously with rendering and is very efficient.
> * In addition, the offset of the point is also related to the shape and expression. These two parameters determine the position of the point before the offset. At the same time, the number of points is large (5023), which makes it difficult to directly learn the offset of each expression point.
> * But we also experimented with adding additional offsets to the expression Gaussians. Our setup is as follows: we extract global features from the driving image, then use the initial positions given by the 3DMM and trainable point features to predict point offsets, and we apply regularization to prevent excessive displacement. The results are shown below.
> * | Method | PSNR↑ | SSIM↑ | LPIPS↓ | CSIM↑ | AED↓ | APD↓ | AKD↓ | CSIM↑ | AED↓ | APD↓ |
> | ---- | ---- | ---- | ---- | ---- | ---- | ---- | ---- | ---- | ---- | ---- |
> | with offsets | **21.93** | 0.814 | 0.138 | 0.797 | 0.121 | 0.168 | 3.732 | 0.584 | 0.263 | 0.294 |
> | Ours | 21.83 | **0.818** | **0.122** | **0.816** | **0.111** | **0.135** | **3.349** | **0.633** | **0.253** | **0.247** |
> * It can be seen that in such a setting, adding offset predictions has little effect or even worse impact on most metrics.
>
> **Would the two sets of Gaussians contract each other during expression?**
> * During the expression (inference) process, the two sets of Gaussians do not contract towards each other. But during training, we use MSE loss to bring the two sets of Gaussians closer together to fully leverage the priors of the 3DMM. So the two sets of Gaussians are close to each other after training.
>
> **Why not bind the lifted Gaussians to the 3DMM vertices and animate them together?**
> * Our primary consideration is efficiency. In our setup, we obtain 175,232 lifted Gaussians and 5,023 expression Gaussians, which require a large matrix to compute their binding relationship. Unlike full-body reconstruction, which involves significant deformations (requiring binding points to a few bones), facial reconstruction doesn't demand such extensive deformations. Therefore, we first attempted to merge the expression Gaussians and enhancement Gaussians directly without binding. The results indicate that this approach is effective for capturing expressions and jaw movements.
> * We have further discussed how we resolved the conflict between the lifting Gaussians and the expression Gaussians caused by not binding in our response to reviewer d3t7.
>
> **Why do we only use the vertices of the 3DMM model?**
> * The vertices are naturally compatible with the positions of the Gaussians in 3DGS. Additionally, 3DMM (FLAME) has 5,023 points, providing sufficient information for head expression control. To integrate 3DMM more effectively into 3DGS, we chose not to use edge and face information.
> * Moreover, the points in 3DMM contain richer information beyond just shape and expression parameters. Therefore, we opted not to redundantly input the shape and expression parameters of 3DMM into the model.
>
> **Side-by-side video comparison with baselines.**
> * Due to rebuttal restrictions, we are unable to provide additional videos for rebuttal. Instead, we have included some consecutive frames in the new supplementary page and highlighted the areas of interest. We will add side-by-side video comparisons in our future open-source code and demo website.
>
> **About the limitations of 3DMM.**
> * We further discussed the impact of 3DMM in our rebuttal to reviewer **65B2**.

---

> > ### Comment · Reviewer_3QbB · 2024-08-12
> > **Thank you**
> >
> > Thank you for the detailed response and additional experiments. I will maintain my initial score of weak accept.

---

> > > ### Author Response · Authors · 2024-08-13
> > >
> > > We thank the reviewer for their thoughtful feedback and for taking the time to review our rebuttal and engage with us. Your comments are important for us. We are pleased we could address your concerns.
> > >
> > > We are happy to address any further concerns.

---

### Official Review · Reviewer_PdVU · 2024-07-12

**Soundness:** 3
**Presentation:** 3
**Contribution:** 3
**Rating:** 5
**Confidence:** 4

**Summary:**

This paper presents "Generalizable and Animatable Gaussian Head Avatar" (GAGA), a method for one-shot animatable head avatar reconstruction using 3D Gaussians. Unlike existing methods that depend on neural radiance fields and require extensive rendering time, GAGA employs a dual-lifting method to generate high-fidelity 3D Gaussians from a single image in a single forward pass. The approach integrates global image features and 3D morphable models to control expressions and achieve real-time reenactment speeds. Experiments demonstrate that GAGA outperforms previous methods in reconstruction quality and expression accuracy.

**Strengths:**

1. The dual-lifting method for reconstructing 3D Gaussians from a single image is novel and effective, allowing for high-fidelity reconstruction.
2. Efficient resource utilization during training and inference makes it practical for real-time applications, which is a significant improvement over existing methods that are often slow.
3. The model can generalize to unseen identities without specific optimizations, broadening its applicability.
4. Experimental results show superior reconstruction quality and expression accuracy performance compared to state-of-the-art methods.

**Weaknesses:**

1. The reliance on 3D morphable models for expression control may limit the model's flexibility.
2. The 3DMM identity and expression parameters are not entirely decoupled, potentially impacting cross-identity reenactment identity consistency.
3. The method may generate less detail for unseen areas and has limitations in controlling regions not modeled by 3DMM, such as hair and tongue.

**Questions:**

1. How sensitive is the model to variations in the input image quality and lighting conditions?
2. How robust is the model when applied to images with significant occlusions or extreme facial expressions?
3. What specific measures were taken to ensure the diversity and quality of the training data?

**Limitations:**

1. The model may produce less detail for unseen areas, which can affect the realism of the generated avatars in dynamic scenes.
2. The expression control is limited by the constraints of the 3DMM, which does not model elements like hair and tongue, leading to potential inaccuracies.
3. There is a slight compromise in identity consistency of cross-identity reenactment due to the partial decoupling of identity and expression parameters in the 3DMM.
4. While the method achieves real-time speeds, the dual-lifting and neural rendering processes add complexity to the implementation and may require significant computational resources for training.

---

> ### Author Rebuttal · Authors · 2024-08-07
>
> Thank you for your positive review and valuable comments. We are pleased to address your concerns in the following sections:
>
> **How sensitive is the model to image quality and lighting conditions?**
>
> * We present more qualitative results with low-quality images or challenging lighting conditions in Fig. 1 of the new supplementary page.
> The reconstructed avatars are inevitably affected by image quality and lighting conditions. For example, avatars reconstructed from blurred images lack details, while those from images with challenging lighting conditions have a fixed lighting condition, such as some shadows on the nose. However, these features also demonstrate that our model can faithfully restore details from the input image and handle scenarios with varying image quality and challenging lighting.
>
> **How robust is the model to significant occlusions or extreme facial expressions?**
>
> * In Fig. 1, we also show the results for inputs with significant occlusions and extreme expressions. For common occlusions such as sunglasses, the model can handle them well. For some uncommon occlusions such as hands, the reconstructed avatar will be affected by the occlusion. For extreme inputs or driving expressions, our method shows good robustness. This shows that our method can produce reasonable results even in extreme cases.
>
> **What specific measures were taken to ensure the diversity and quality of the training data?**
>
> * We constructed our training data from the VFHQ dataset, a talking head video dataset containing 15,204 video clips. Despite the repeated identities, the dataset still has a large number of unique identities, ensuring sufficient identity diversity in our training data.
> To ensure diversity in expressions and head poses between frames, we extracted frames from the videos. As described in Sections 4.1 and A.1, we uniformly sampled 25-75 frames from each video based on its length. This temporally sparse sampling ensures that different frames exhibit as much variation in expressions and poses as possible.
>
> * For the ground truth 3DMM parameters in our dataset, we adopted the implementation from GPAvatar [Chu et al., 2024]. This implementation integrates state-of-the-art 3DMM estimators (MICA, EMOCA), resulting in high-quality 3DMM labels.
>
> **Implementation complexity and computational resources for training.**
>
> * Our method is also highly efficient in training. As shown in the table below, our training costs are lower than those of the baseline methods. In addition, as a generalizable method, our model only needs to be trained once and does not require fine-tuning during inference.
> In the future, we plan to open-source our code, including implementations of all components, as well as our pre-trained models. We hope this will provide an easy-to-use and solid baseline for future research.
>
> * | Method |StyleHeat | ROME | OTAvatar | HideNeRF | GOHA | CVTHead | GPAvatar | Real3D | P4D | P4D-v2 | Ours|
> | ----------- | ----------- | ----------- | ----------- | ----------- | ----------- | ----------- | ----------- | ----------- | ----------- | ----------- | ----------- |
> | **Time(GPU Hours)**  |166           |    -     | 192           | 1667           | 768      | 1200        | 50           | 1424     | 672  | 1536      |  **46** |
>
> * Among them, ROME did not provide training details and code. StyleHeat, HideNeRF, and CVTHead did not release official training time, so we conducted limited training and estimated the total time for these methods. All times tested on / reported on /or converted to A100.
>
> **About the limitations of 3DMM.**
> * We further discussed the impact of 3DMM in our rebuttal to reviewer **65B2**.

---

> > ### Comment · Reviewer_PdVU · 2024-08-13
> > **Thanks for the rebuttal.**
> >
> > Thank you for the rebuttal. After reading the comments from other reviewers and the reponses from authors, I would keep my rating as borderline accept.

---

> > > ### Author Response · Authors · 2024-08-13
> > >
> > > Thank you for your thoughtful feedback and for taking the time to review our rebuttal and engage with us. Your comments are important for us. We are pleased we could address your concerns.
> > >
> > > We are happy to address any further concerns.

---

### Author Rebuttal · Authors · 2024-08-07

Firstly, we would like to thank all the reviewers for their thorough review and valuable suggestions. We summarize the issues raised and indicate where we address them. Additionally, we provide some new visual results in the supplementary page.

We answered the following questions in our response to **PdVU**:
* Robustness of the model to image quality, lighting condition, occlusion, and extreme expression. (**PdVU**)
* How we ensure the diversity of training data. (**PdVU**)
* The complexity of model implementation and the resource consumption during the training process. (**PdVU**)

We answered the following questions in our response to **3QbB**:
* Why don’t we predict offsets for expression Gaussians? (**3QbB**)
* Do the lifted Gaussians and expression Gaussians contract each other during expression? (**3QbB**)
* Why don’t we bind the lifted Gaussians to the 3DMM expression Gaussians? (**3QbB**)
* Why do we use only 3DMM vertices? (**3QbB**)

We answered the following questions in our response to **65B2**:
* Ablation study of removing the reconstruction branches (**65B2**)
* Explanation of quantitative results and over-smoothed qualitative results. (**65B2**)
* Discussion on the impact of 3DMM accuracy on model performance. (**65B2**, **PdVU**)
* There should be some failure cases to show the limitations of the work. (**65B2**)

We answered the following questions in our response to **d3t7**:
* Further explanation of the model design, including how we model dynamic expressions. (**d3t7**)
* How we resolve conflicts between the two sets of Gaussian points. (**d3t7**, **3QbB**)
* Further discussion of the novelty of our method. (**d3t7**)
* Details on how we conduct our evaluation and comparison. (**d3t7**)

We also show more results in the new supplementary page:
* Side-by-side video frames comparison results and qualitative results with highlights. (**3QbB**, **d3t7**) Due to page limitations we selected several of the most competitive baseline methods to show video frames.

---

### Comment · Area_Chair_GTXy · 2024-08-12
**Author Rebuttal**

Dear Reviewers,

The authors have provided detailed responses to the questions that you raised. I strongly encourage you to please read the authors' responses and acknowledge that you have read them before the AOE 11:59 PM (Aug 13, 2024).

Best,
AC

---

### Comment · Area_Chair_GTXy · 2024-08-13
**Response to Authors**

Dear Reviewer d3t7,

The authors have provided a response to the questions you raised in your rebuttal. The discussion period with them is soon coming to a close, in less than 24 hours at 11:59 PM AOE Aug 13, 2024. It would be great if you could acknowledge that you've read the authors response, engage in any further discussion with them and update your score if needed.

Best,
AC

---

### Decision · Program_Chairs · 2024-09-25

**Decision:**

Accept (poster)

**Comment:**

This paper proposes a method for one-shot 3D reconstruction of faces from single input images along with their animation using input FLAME expression parameters. The authors propose a dual Gaussian lifting framework to lift a static head and the expression coefficients into a Gaussian representation. Additionally for the static branch, a backwards and forwards lifting approach is proposed, which is novel. A neural render is further proposed to resolve any discrepancies between the dynamic and static branches. The proposed approach also relies heavily on the parametric 3DMM model for regularizing 3D reconstruction. The proposed algorithm is compared to several existing state of the art baselines on two benchmark datasets and shown to be better in terms of reconstruction accuracy and expression preservation, but not SOTA in terms of identity preservation. By using Gaussian splatting the proposed method is also fast in terms of training and inference times versus the existing approaches. Lastly, without requiring test-time fine-tuning to new subjects the method generalizes out of the box to unseen faces.

The reviewers appreciated the novelty of the proposed approach in being the first to employ Gaussian splatting for generalized one-shot animatable avatar creation, its speed, generalization ability and the quality of the results. However, reviewers raised concerns about the heavy reliance on a 3DMM, which does not allow for parts of the face that aren't mapped to the 3DMM (hair and tongue) to be animatable. Requiring FLAME fitting, also makes identity and expression parameters to not be disentangled fully. Reviewers also raised concerns about the design choice of employing the dual independent dynamic and static branches both if which predict Gaussians, which could lead to differing shapes and hence is not an intuitive design.

All things considered, the AC feels that the proposed approach is a step in the right direction towards employing 3D Gaussians for generalized animatable 3D head avatar creation.While not perfect, it would form a strong baseline for future work to build upon and encourage further exploration and improvements w.r.t. moving away on the reliance on 3DMM. The authors have also promised to release their code. Hence, the AC leans towards accepting this work.